# Microbial reduction of metal-organic frameworks enables synergistic chromium removal

Sarah K. Springthorpe [1], Christopher M. Dundas [2] & Benjamin K. Keitz [2]*

Redox interactions between electroactive bacteria and inorganic materials underpin many emerging technologies, but commonly used materials (e.g., metal oxides) suffer from limited tunability and can be challenging to characterize. In contrast, metal-organic frameworks exhibit well-defined structures, large surface areas, and extensive chemical tunability, but their utility as microbial substrates has not been examined. Here, we report that metal-organic frameworks can support the growth of the metal-respiring bacterium *Shewanella oneidensis*, specifically through the reduction of Fe(III). In a practical application, we show that cultures containing *S. oneidensis* and reduced metal-organic frameworks can remediate lethal concentrations of Cr(VI) over multiple cycles, and that pollutant removal exceeds the performance of either component in isolation or bio-reduced iron oxides. Our results demonstrate that frameworks can serve as growth substrates and suggest that they may offer an alternative to metal oxides in applications seeking to combine the advantages of bacterial metabolism and synthetic materials.

[1] Department of Chemistry, University of Texas at Austin, 200 East Dean Keeton Boulevard C0400, Austin, TX 78712, USA. [2] McKetta Department of Chemical Engineering, University of Texas at Austin, 200 East Dean Keeton Boulevard C0400, Austin, TX 78712, USA. *email: keitz@utexas.edu

Redox-active minerals support the growth of microorganisms capable of extracellular electron transport in anaerobic environments[1]. These microbe–mineral interactions play a key role in biogeochemical processes, including the nitrogen and carbon cycles[2,3], as well as nascent technologies like microbial fuel cells[4] and bioremediation[5]. For many of these applications, iron oxides are commonly employed as low-cost and widely available substrates for microbial reduction. However, iron oxides are significantly limited in structural diversity and only a fraction can be prepared under laboratory conditions[6]. Moreover, even relatively accessible structures, such as ferrihydrite (FeOOH·0.4H$_2$O), are recalcitrant to structural characterization[6]. As such, synthetic materials may offer opportunities to better leverage microbial metabolism and improve applications dependent upon both microbial physiology and material characteristics.

One emerging alternative to metal oxides are metal-organic frameworks. Relative to metal oxides, these materials generally exhibit higher surface areas, superior crystallinity, and can possess oxide-like redox-activity[7]. Metal-organic frameworks also facilitate the development of highly tunable structure-function relationships since properties such as pore size, linker identity, and metal node can be varied independently of one another[8]. As a result, they have received significant interest across several fields, including gas separations[9], catalysis[10], drug delivery[11], and environmental remediation[12]. Despite their advantages over metal oxides and potential presence in nature[13], metal-organic frameworks have not been examined as substrates for the growth of metal-reducing or -oxidizing microorganisms. Owing to their high-surface areas and redox-active nature, we hypothesize that metal-organic frameworks can support microbial growth and potentially augment biological/material applications that rely upon biologically driven redox transformations.

Here, we test whether select iron-based metal-organic frameworks can serve as respiratory electron acceptors for the metal-reducing bacterium *Shewanella oneidensis* MR-1 (Fig. 1). The frameworks Fe-BTC (BTC = 1,3,5-benzenetricarboxylate), Fe$_3$O(BTC)$_2$(OH)•nH$_2$O (MIL-100), and Fe$_3$O[(C$_2$H$_2$(CO$_2$)$_2$]$_3$(OH)•nH$_2$O (MIL-88A) are used for our growth and reduction assays, as their nodes are a well-precedented *S. oneidensis* substrate (namely, Fe(III)), the frameworks are water-stable, and they span a range of morphologies. We demonstrate that bacterial growth and Fe(III) reduction can occur on metal-organic frameworks in the absence of material toxicity, and that reduction occurs using known *S. oneidensis* metabolic and protein (MtrCAB) pathways (Fig. 1a). Two frameworks (MIL-100 and Fe-BTC) exhibit accelerated reduction rates relative to a model iron oxide

(ferrihydrite), which we attribute to the higher surface areas of the metal-organic frameworks.

Applying this microbe-material pairing, we show that *S. oneidensis* and metal-organic frameworks synergistically facilitate bioremediation of Cr(VI), a known environmental and carcinogenic pollutant. Microbial reduction of framework-bound Fe(III), especially in MIL-100, generates high levels of redox-active Fe(II) that increases Cr(VI) reduction and its adsorption to frameworks. Notably, Cr(VI) removal rates and capacities drastically exceed those of biotic metal oxides, abiotic frameworks, and *S. oneidensis* alone. When operated together, a cycling treatment scheme is enabled, whereby *S. oneidensis* continuously regenerates reactive Fe(II) that can treat multiple Cr(VI) additions. Under our conditions, we found that MIL-100 protects the bacteria from lethal challenges of Cr(VI) and that tens of treatment cycles are possible. Overall, our results highlight the ability of metal-organic frameworks to support bacterial growth and reduction in a structure dependent manner and demonstrate how the advantages of these materials can be co-opted for pollutant adsorption when coupled to bacterial metabolism.

## Results

**Abiotic stability of metal-organic frameworks.** Prior to using the materials as microbial substrates, we examined the stability of metal-organic frameworks under abiotic culture conditions. We chose the Fe(III)-containing metal-organic frameworks MIL-100, Fe-BTC, and MIL-88A due to their reported aqueous stability, range of crystallinities, and range of surface areas[14,15]. To determine if the materials were stable in typical culture conditions, each metal-organic framework (initial Fe(III) concentration set to 15 mM) was suspended in *Shewanella* Basal Medium (SBM) containing 20 mM lactate and anaerobically stored at 30 °C for 48 h. Our SBM formulation was buffered with 100 mM HEPES and contained 0.05% casamino acids and 1x Wolfe's Mineral Solution (ATCC). After 48 h, powder X-ray diffraction (PXRD) patterns were collected and the leaching of potential microbial substrates, namely Fe(III) for all three metal-organic frameworks and fumarate in MIL-88A, was also monitored. MIL-100 and Fe-BTC maintained structural integrity, as indicated by agreement between the as-synthesized and exposed patterns. On the other hand, MIL-88A exhibited a slight, but detectable, change in its PXRD pattern (Supplementary Fig. 1a). Furthermore, none of the materials showed significant leaching of Fe(III) or fumarate (Supplementary Fig. 1b, c). These results indicate that the metal-organic frameworks were stable under our culture conditions and could be used as insoluble substrates for *S. oneidensis*.

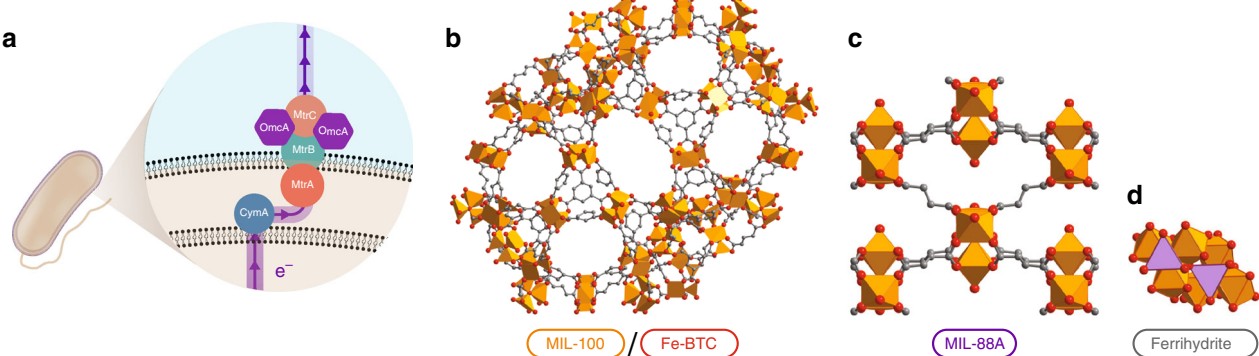

**Fig. 1** *S. oneidensis* MR-1 metal reduction pathway and metal-organic framework structure. **a** The MtrCAB pathway by which *S. oneidensis* reduces both soluble and insoluble metal species[1]. **b**–**d** The crystal structure of MIL-100/Fe-BTC[57,58] (**b**), MIL-88A[59] (**c**), and ferrihydrite[44] (**d**). Fe-BTC does not have a known crystal structure (due to its amorphous nature), but it is analogous to MIL-100

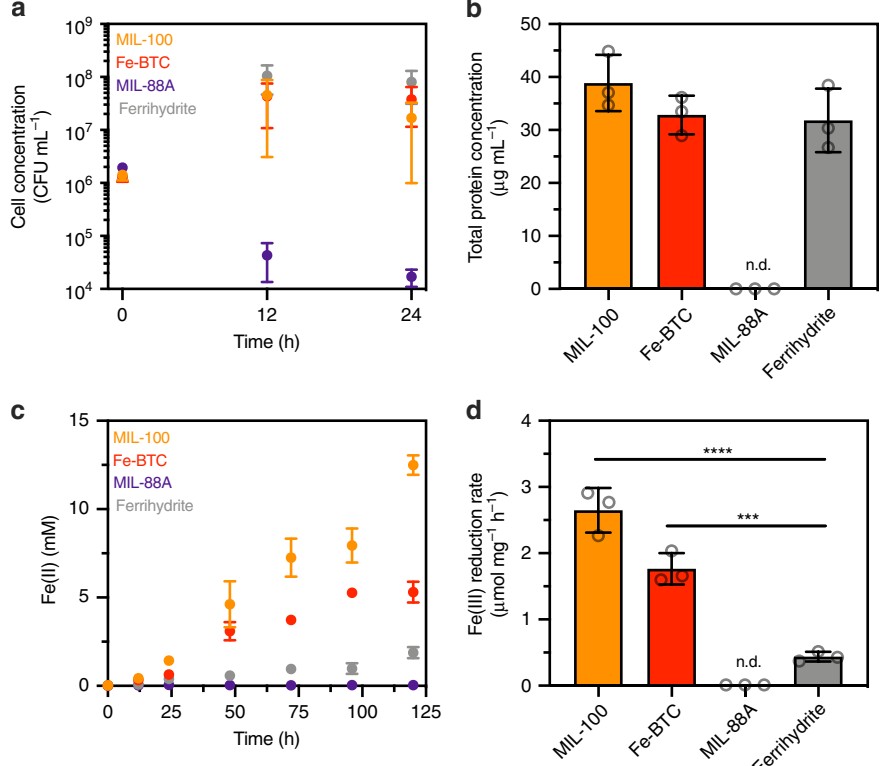

**Fig. 2** Growth of *S. oneidensis* MR-1 and reduction of metal-organic frameworks. **a** CFU counts of *S. oneidensis* MR-1 (inoculating $OD_{600} = 0.002$) grown on MIL-100, Fe-BTC, MIL-88A, and ferrihydrite over 24 h. Data show mean ± S.E.M. for three independent biological replicates. A two-tailed *t*-test was performed using the MIL-88A data between time 0 h and 24 h, $P = 0.028$. **b** Total protein concentration of *S. oneidensis* MR-1 (inoculating $OD_{600} = 0.002$) grown on MIL-100, Fe-BTC, MIL-88A, and ferrihydrite after 120 h. Data shown are mean ± S.D. for three independent biological replicates. **c** Total Fe(II) concentrations in suspensions containing *S. oneidensis* MR-1 (inoculating $OD_{600} = 0.002$) and either MIL-100, Fe-BTC, MIL-88A, or ferrihydrite over 120 h. Data shown are mean ± S.D. for three independent biological replicates. **d** Total protein normalized Fe(III) reduction rates for *S. oneidensis* (inoculating $OD_{600} = 0.002$) MR-1 grown on either MIL-100, Fe-BTC, MIL-88A, or ferrihydrite as measured at 120 h. n.d. indicates that the rate could not be determined due to lack of detectable biomass. Data shown are mean ± S.D. for three independent biological replicates. Figures **b**–**d** were collected in the same experiment. For panel **d**, statistical comparisons were made to ferrihydrite using a one-way ANOVA with Dunnett's post-test, ***$P = 0.001$, ****$P < 0.0001$. Source data for **a**–**d** are provided as a Source Data file

**Frameworks support microbial growth and reduction**. We first examined whether *S. oneidensis* MR-1 could use iron-based metal-organic frameworks as respiratory substrates for cell growth. In addition to the frameworks, MIL-100, Fe-BTC, and MIL-88A, we utilized the *Shewanella* growth substrate ferrihydrite in order to draw comparisons with iron oxides. To monitor the growth of *S. oneidensis*, each material was suspended in SBM containing lactate as the carbon source and inoculated with anaerobically pregrown cells. Colony-forming units (CFU) from plated dilutions of each material-bacteria suspension were used to measure cell concentration. With MIL-100, Fe-BTC, and ferrihydrite suspensions, we observed comparable increases in cell concentration after 12 h that stayed constant up to 24 h (Fig. 2a). Concomitant Fe(III) reduction was also observed with these three materials (Supplementary Fig. 2). In contrast, cell concentration in the MIL-88A suspension significantly decreased over a 24-h period ($P = 0.028$, two-tailed *t*-test), while Fe(II) levels slightly increased. As a control, we measured cell concentration of *S. oneidensis* MR-1 in electron acceptor-free medium and found that, although CFU counts slightly increased over this same timescale, they were still an order of magnitude lower than suspensions containing MIL-100, Fe-BTC, and ferrihydrite (Supplementary Fig. 3). Additionally, we incubated *Escherichia coli* MG1655 in lactate-supplemented media with and without MIL-88A. In both cases, CFU counts increased to comparable levels, indicating MIL-88A does not impede *E. coli* growth. Thus, the

decrease in viable CFU counts in *S. oneidensis*-MIL-88A suspensions is not due to inability of *S. oneidensis* to respire and suggests MIL-88A exhibits *Shewanella*-specific toxicity. We posited this toxicity might arise due to MIL-88A particle morphology. Indeed, scanning electron micrographs (SEM) revealed that MIL-88A formed small needle-like particles whereas MIL-100, Fe-BTC, and ferrihydrite generated large, morphologically similar particle aggregates (Supplementary Fig. 4). Overall, the frameworks MIL-100 and Fe-BTC facilitated increases in cell growth that were associated with simultaneous Fe(III) reduction, similar to ferrihydrite.

As CFU counts on insoluble materials can violate the assumption that one bacterium results in one CFU, we sought to confirm cell growth via other biomass measurements. Increases in cellular DNA were measured using the fluorescent dye, Syto™ 9 (ThermoFisher Scientific). The fluorescent dye SYPRO™ Ruby (ThermoFisher Scientific) was also used to monitor increases in extracellular biofilm matrix proteins. This non-specific dye is known to stain a large variety of proteins, including glycoproteins, phosphoproteins, lipoproteins, calcium binding proteins, and fibrillar proteins. Starting with a tenfold higher starting concentration of *S. oneidensis* MR-1 compared to our CFU counts (to improve signal-to-noise), we in parallel measured Fe(II) levels, Syto™ 9 fluorescence, and SYPRO™ Ruby fluorescence over 48 h for MIL-100 and ferrihydrite suspensions. For each bacteria-material suspension, Syto™ 9 signal increased and rapidly

plateaued to relatively similar extents after ~8 h (Supplementary Fig. 5a). Likewise, the SYPRO™ Ruby signals for biotic material suspensions increased at comparable rates (Supplementary Fig. 5b). Altogether, these fluorescence results corroborate the similar increases in CFU counts for *S. oneidensis* grown on MIL-100 and ferrihydrite.

Finally, we quantified total protein levels after material reduction using the Bradford assay. With the exception of MIL-88A, all material suspensions exhibited similar increases in protein content after 120 h and were measured as ca. 30–40 μg mL$^{-1}$ (Fig. 2b). This yield is comparable to previously reported stationary-phase protein levels for *S. oneidensis* MR-1[16]. Protein levels in the MIL-88A suspension were below the assay's limit of detection, which indicated a lack of appreciable biomass accumulation and further confirmed the measured decreases in CFU counts. Based on all our biomass/CFU results, in the absence of material toxicity, biomass accumulation appears relatively invariant to the framework used and comparable to growth with a model iron oxide (ferrihydrite). Our results are also in agreement with previous *Shewanella* growth studies on soluble/insoluble iron substrates, which showed that cell yields were similar regardless of electron acceptor used[17].

Simultaneous monitoring of Fe(II) levels over time and endpoint biomass measurements for each material-bacteria suspension enabled the calculation of biomass-normalized Fe (III) reduction rates. Across all abiotic material suspensions, Fe (II) levels remained low and did not increase in control experiments, indicating that increases in Fe(II) were due to cellular activity. In contrast, each biotic material suspension showed distinct extents of Fe(III) reduction over time (Fig. 2c). After 120 h, both MIL-100 and Fe-BTC exhibited similar biomass-normalized rates of reduction that were significantly higher from that of ferrihydrite (Fig. 2d). Comparable trends in reduction rate were observed from biomass and Fe(II) levels measured after 48 h (Supplementary Fig. 6). At all timescales, the Fe(III) reduction rate for MIL-88A could not be determined due to the lack of appreciable Fe(II) and biomass. Encouraged by the drastic differences in raw Fe(II) kinetics and biomass-normalized Fe(III) reduction rates with each framework and ferrihydrite, we further examined the biological and material effects influencing framework reduction.

**Genetic and metabolic effects on framework reduction**. Towards understanding the mechanism of metal-organic framework reduction by *S. oneidensis*, we assessed the importance of key genetic and metabolic factors that are well established for metal oxide reduction. We tested whether Fe(III) reduction is a result of active cellular metabolism by comparing Fe(II) levels across material suspensions containing either healthy or metabolically impaired *S. oneidensis* MR-1 cells. In agreement with our observed increases in CFU counts and biomass on MIL-100 and Fe-BTC, heat-killed cells and cell lysate did not yield appreciable Fe(III) reduction on these materials (Fig. 3a). These results imply that growth and metabolic activity are necessary for Fe(III) reduction in the tested frameworks. In our experiments, lactate was presumed to be the primary electron donor of our medium formulation. Indeed, higher levels of Fe(III) reduction occurred in the presence of 20 mM lactate relative to formulations where it was omitted (Fig. 3b). However, even without lactate, Fe(II) levels were markedly higher in biotic samples over abiotic, suggesting other electron donors (e.g., chamber atmosphere H$_2$) were metabolically accessible. In lactate-supplemented medium, metabolite measurements revealed that lactate consumption generally followed the extent of Fe(III) reduction for *S. oneidensis* MR-1 cultures containing either MIL-100, Fe-BTC, or ferrihydrite

(Supplementary Fig. 7a). Additionally, metabolic intermediates associated with lactate catabolism/anaerobic respiration were generated in both framework- and ferrihydrite-containing cultures (i.e., pyruvate and acetate)[18]. Collectively, these results indicate that *S. oneidensis* reduces Fe(III) in MIL-100, Fe-BTC, and iron oxides using similar metabolic pathways (Supplementary Fig. 7b).

We next attempted to identify the direct contributors to Fe(III) reduction by *S. oneidensis* and determine how they compare to previously reported iron oxide reduction pathways. In *S. oneidensis* MR-1, catabolic electron flux is linked to the respiration of extracellular metal oxides via the MtrCAB pathway[19]. Specifically, the outer-membrane cytochromes MtrC and OmcA enable the terminal extracellular electron transfer step onto both soluble and insoluble metal species[20]. Without these proteins, *S. oneidensis* shows attenuated respiration onto iron and graphene oxides[21]. Consistent with these results, an outer-membrane cytochrome deficient strain, Δ*mtrC*Δ*omcA*, showed diminished Fe(III) reduction and biomass accumulation when cultured with MIL-100 or Fe-BTC, relative to MR-1 (Fig. 3c). *E. coli* MG1655, which does not possess a homolog to MtrC or OmcA and lacks electroactivity, was also unable to reduce Fe(III) in these materials (Supplementary Fig. 8). In addition to direct cytochrome-based reduction, *S. oneidensis* can use soluble redox shuttles (flavins) to indirectly transfer reducing equivalents from the cell to metal oxides, with exogenously supplied flavins accelerating the rate of Fe(III) reduction[22]. Flavins bound to extracellular cytochromes can also potentially mediate extracellular electron transfer[23]. To examine the role of flavins, we cultured a *S. oneidensis* mutant that exhibits diminished flavin secretion (Δ*bfe*) with MIL-100 and Fe-BTC. While this mutant was previously reported as having Fe(III) reduction defects when grown on ferrihydrite[24], we did not observe a statistically different biomass-normalized Fe(III) reduction rate when grown on metal-organic frameworks relative to the wild-type strain (MR-1) (Fig. 3c). Similarly, we found that supplementation of 1 μM or 10 μM riboflavin to *S. oneidensis* MR-1 cultures containing MIL-100 or Fe-BTC led to slight but statistically non-significant increases in Fe(III) reduction rates relative to non-supplemented controls (Supplementary Fig. 9), indicating *S. oneidensis* MR-1 may be accessing soluble iron[24]. Thus, our data confirms that *S. oneidensis*-based Fe(III) reduction in metal-organic frameworks follows similar metabolic and protein pathways as iron oxides, and is primarily mediated by extracellular cytochromes.

**Stability of frameworks in the presence of *S. oneidensis***. Next, we asked if Fe(III) reduction and metabolic activity from *S. oneidensis* negatively impacted solution stability of the frameworks. Microbial reduction of iron oxides increases dissolution of oxide-bound Fe[25] and we hypothesized that similar dissolution may occur with framework-bound Fe. Additionally, *S. oneidensis* may only be able to reduce soluble Fe(III), which creates a driving force for further leaching and framework destabilization. To probe both of these scenarios, we examined post-reduction PXRD patterns of each material. As previously mentioned, MIL-100 and Fe-BTC maintained structural integrity while MIL-88A exhibited a slight, but detectable, change in structure under abiotic culture conditions. However, post-reduction PXRD patterns showed that MIL-88A eventually decomposed under biotic conditions after 48 h (Fig. 4a). In contrast, characteristic MIL-100 and Fe-BTC peaks were discernable after exposure to the bacteria, indicating that these materials remain structurally intact and crystalline over the course of our experiments. Additionally, we qualitatively observed that MIL-100 and Fe-BTC changed from orange to gray/black following reduction by *S. oneidensis*, suggesting Fe(II)

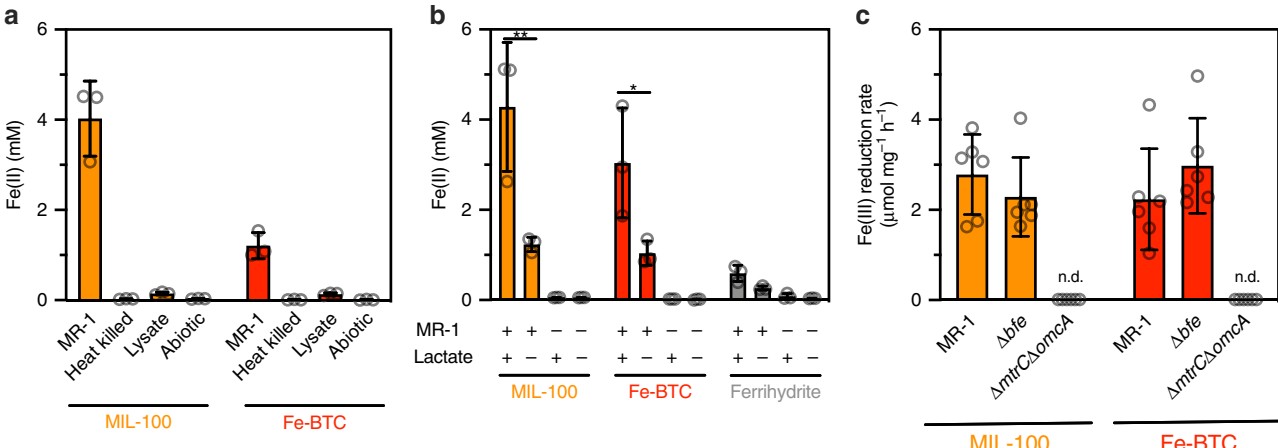

**Fig. 3** Genetic and metabolic effects on reduction. **a** Fe(II) concentrations after 48 h in cultures containing either heat-killed, lysed, or whole-cell inocula of *S. oneidensis* MR-1 (inoculating $OD_{600} = 0.002$). Data shown are mean ± S.D. for three independent biological replicates. **b** Fe(II) concentrations after 48 h for cultures of *S. oneidensis* MR-1 (inoculating $OD_{600} = 0.02$) with 0 mM lactate or 20 mM lactate. + indicates the presence of either *S. oneidensis* MR-1 or lactate, while – indicates the absence of the components. Data shown are mean ± S.D. for three independent biological replicates. Three abiotic replicates are also shown. Statistical comparisons were made between lactate+ and lactate– biotic samples using two-way ANOVA and Sidak's post-test. *$P = 0.0255$, **$P = 0.0014$. **c** Total protein normalized Fe(III) reduction rate of *S. oneidensis* MR-1 and knockout strains Δ*bfe* and Δ*mtrC*Δ*omcA* after 48 h. n.d. indicates that the rate could not be determined due to lack of detectable biomass. Data shown are mean ± S.D. for six independent biological replicates. Source data for **a**–**c** are provided as a Source Data file

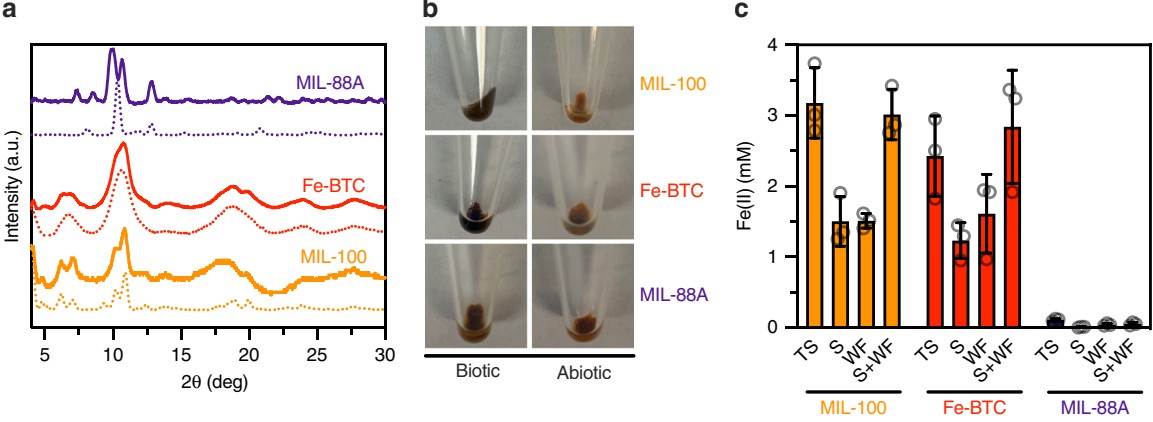

**Fig. 4** Stability of metal-organic frameworks in the presence of *S. oneidensis*. **a** As-synthesized (dashed lines) and post-reduction (solid lines) PXRD patterns of MIL-100, Fe-BTC, and MIL-88A. **b** Images of biotic and abiotic treated MIL-100, Fe-BTC, and MIL-88A after 48 h. **c** Fe(II) concentrations as measured in the total suspension (TS), isolated supernatant (S), and isolated/washed framework (WF). The sums of the isolated supernatant and isolated/ washed framework are also shown (S + WF). Data shown are mean ± S.D. for three independent biological replicates. Source data for **a** and **c** are provided as a Source Data file

was contained in the framework (Fig. 4b). Similar color changes have previously been noticed with Fe-BTC when the framework was exposed to chemically reducing conditions[26]. We quantitatively confirmed that Fe(II) was contained within each framework by measuring Fe(II) in pelleted/washed samples of biotic MIL-100 and Fe-BTC (Fig. 4c). Nonetheless, ca. 50% of the total reduced Fe in framework suspensions was soluble Fe(II) after 48 h of microbial culture. Similar to observations with iron oxides that are partially solubilized upon microbial reduction, our results indicate that *S. oneidensis* can both directly reduce Fe(III) in MIL-100 and Fe-BTC and partially remove Fe(II/III) from the framework structure while maintaining framework stability[25].

**Cr(VI) adsorption by microbially reduced frameworks.** Having determined that *S. oneidensis* electroactive metabolism can interface with structurally intact metal-organic frameworks, we sought

to harness this microbe-material pairing for environmental remediation. Detoxification and separation of environmental contaminants often leverages redox-based transformations, and many schemes are enabled by the release of reactive Fe from iron oxides. In these systems, generated Fe(II) can reductively precipitate oxidized organics and transition metals[27]. Likewise, electroactive bacteria can be used in bioreactors to directly reduce these pollutants[28]. As a result, metal-reducing bacteria, acting in conjunction with iron oxides, have shown promise for the bioremediation of redox-controlled toxins[29].

Alternatively, high-surface area inorganic adsorbents, including activated carbons and metal-organic frameworks, can be used for the remediation of environmental pollutants. For example, some metal-organic frameworks have been used for Cr(VI) adsorption, but generally show slow adsorption kinetics and limited capacities[30–32]. A more recent study found that Cr(VI) adsorption to a metal-organic framework could be dramatically improved by

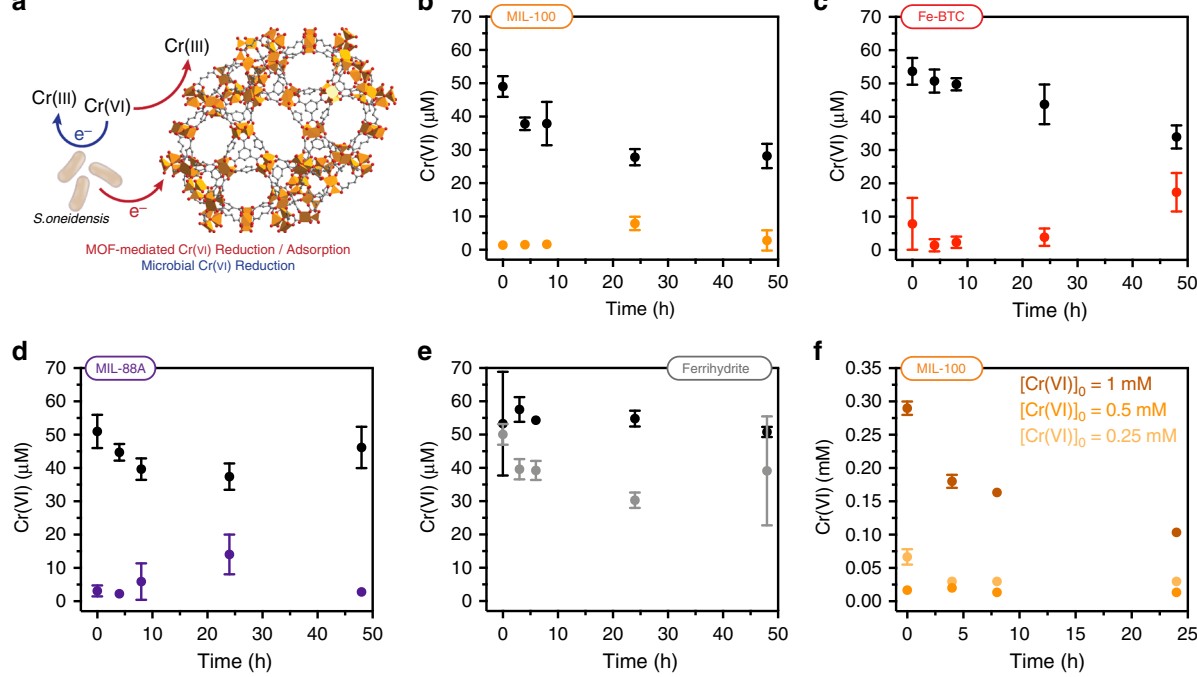

**Fig. 5** Cr(VI) removal in reduced metal-organic frameworks. **a** Generalized scheme for Cr(VI) reduction by both *S. oneidensis* MR-1 and a reduced metal-organic framework. **b–e** Cr(VI) removal by *S. oneidensis* MR-1-reduced **b** MIL-100, **c** Fe-BTC, **d** MIL-88A, and **e** ferrihydrite when challenged with 70 μM Cr(VI). Abiotic samples for each material are shown in black. **f** Cr(VI) removal by *S. oneidensis* MR-1-reduced MIL-100 when challenged with 0.25 mM, 0.5 mM, and 1 mM Cr(VI). All data shown are mean ± S.D. for three independent replicates. Source data for **b–f** are provided as a Source Data file

adding exogenous Fe(II)[33]. This result implies that, similar to the case with iron oxides, the presence of Fe(II) can enhance reductive adsorption of Cr(VI). Drawing influence from these various redox-driven and adsorptive treatments, we hypothesized that microbial reduction of Fe(III)-containing frameworks could exhibit a synergistic effect on Cr(VI) removal. Specifically, we predicted that the reduction of Cr(VI) to Cr(III) by *S. oneidensis* and microbially generated Fe(II) would promote Cr adsorption into the framework (Fig. 5a).

In initial experiments, we grew *S. oneidensis* MR-1 on each metal-organic framework for 24 h and then challenged the culture with a single dose of Cr(VI). At initial Cr(VI) concentrations of 70 μM, all frameworks showed immediate removal of soluble Cr(VI) below the detection limit of the diphenylcarbazide (DPC) assay (Fig. 5b–d). Abiotic controls and biotic ferrihydrite (Fig. 5e) showed minimal and slow Cr(VI) adsorption. In the case of MIL-100 and Fe-BTC, we found that Fe(III) reduction continued to take place after Cr(VI) addition, indicating that *S. oneidensis* was still viable under these conditions. Even though MIL-88A showed minimal Fe(III) reduction in the presence of *S. oneidensis* MR-1, the basal Fe(II) levels still reduced the relatively small initial dose of Cr(VI). These results are a preliminary indication that there is a synergistic interaction between the frameworks and *S. oneidensis* that accelerates Cr(VI) reduction and adsorption.

To understand the range of Cr(VI) concentration that could be treated with our system, we challenged *S. oneidensis* MR-1 growing on MIL-100 with increasing concentrations of Cr(VI) up to 1 mM (Fig. 5f). After dosing with 0.5 mM Cr(VI), *S. oneidensis* MR-1 and MIL-100 reduced all detectable Cr(VI) within minutes. Under these conditions, the total solution concentration of Cr, as measured by inductively coupled plasma-mass spectrometry (ICP-MS), fell below 10 ppm within minutes and reached ~1 ppm after 24 h (Supplementary Fig. 10). Furthermore, Fe(III) reduction continued to take place after Cr(VI) addition, suggesting that

*S. oneidensis* can survive normally toxic Cr(VI) concentrations in the presence of MIL-100[34]. At 1 mM Cr(VI), *S. oneidensis* MR-1 on MIL-100 showed almost complete reduction, but over a much longer time period, indicating that this concentration is approaching the limit of toxicity for our system.

Since our system allows for both bio- and material-based mechanisms of Cr(VI) removal, we sought to disentangle the role of each in reduction/adsorption of the pollutant. As mentioned above, *S. oneidensis* can directly reduce Cr(VI) through the MtrCAB pathway. Alternatively, *S. oneidensis* may indirectly reduce Cr(VI), with bacterially generated Fe(II) from MIL-100 acting as the mediator. Overall, we found that biotic samples in the absence of MIL-100 showed considerably slower Cr(VI) reduction kinetics and were more sensitive to the initial Cr(VI) dose (Supplementary Fig. 11a). These results are consistent with previous reports indicating that 200 μM Cr(VI) is the upper limit for *S. oneidensis* MR-1 survivability[34]. We also measured Cr(VI) reduction/adsorption kinetics for *S. oneidensis* grown on ferrihydrite and found, even at low initial Cr(VI) concentrations, biotic and abiotic ferrihydrite samples exhibited slower adsorption kinetics and capacities relative to all frameworks tested (Fig. 5). Additionally, after *S. oneidensis* had reduced appreciable amounts of Fe(III) in ferrihydrite, there was not a corresponding decrease in Cr(VI), as would have been expected if Fe(II) alone was responsible for Cr(VI) reduction. We further tested the influence of Fe(II) by adding soluble $FeCl_2 \cdot 4H_2O$ to abiotic/material-free Cr(VI) solutions. While we observed a dose dependent increase in Cr(VI) reduction with addition of Fe(II), Cr(VI) reduction rates decreased relative to biotic/material samples and no precipitate was observed (Supplementary Fig. 11b). Altogether, our results support a synergistic interaction between *S. oneidensis* MR-1 and MIL-100 that enhances Cr(VI) reduction/adsorption kinetics and capacity relative to material-free systems, abiotic frameworks, and *S. oneidensis* growing on iron oxides.

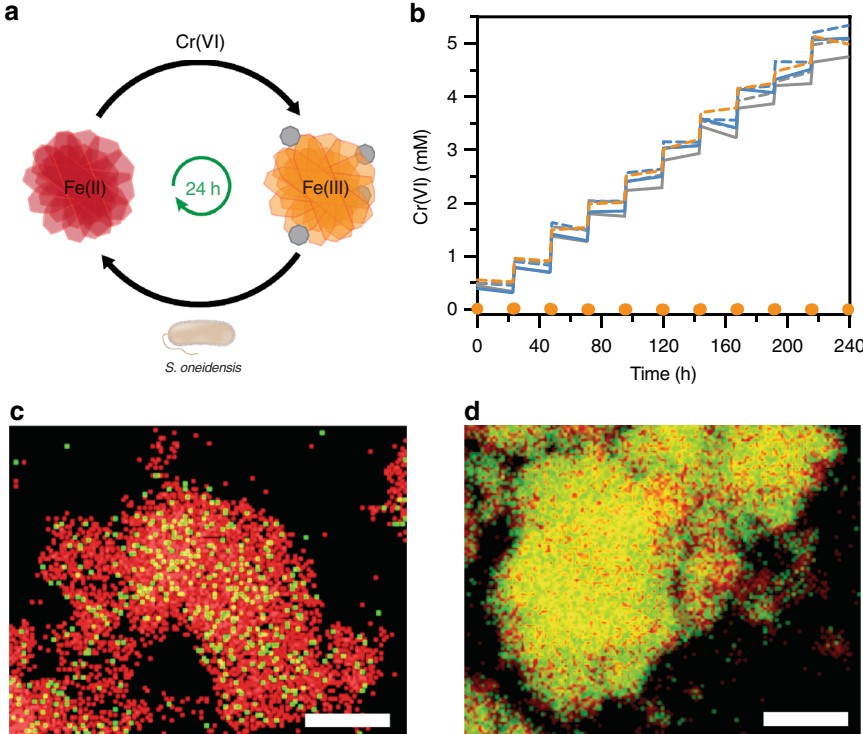

**Fig. 6** Cr(VI) cycling with reduced metal-organic frameworks. **a** Generalized scheme for Cr(VI) cycling where *S. oneidensis* regenerates Fe(II) following Cr (VI) additions. **b** Cr(VI) removal in *S. oneidensis* MR-1-reduced MIL-100 (orange), ferrihydrite (gray) and fumarate (blue) when challenged with ten cycles of 0.5 mM Cr(VI). Abiotic samples are shown as dashed lines. **c**, **d** EDX element mapping of (**c**) abiotic cycled MIL-100 and (**d**) *S. oneidensis* MR-1-reduced and cycled MIL-100. Fe is shown in red and Cr is shown in green. Scale bars are 1 μm. Data show mean ± S.D. for three independent replicates. Source data for **b** are provided as a Source Data file

**Cycles of Cr(VI) removal with *S. oneidensis*-reduced MIL-100.** The majority of materials for Cr(VI) adsorption require a regeneration step using pH adjustments after saturation[35]. Additionally, iron-based treatments require Fe(II) regeneration since Fe(III)-Cr aggregates form when Fe(II) reacts with Cr(VI). Based on our initial Cr(VI) adsorption experiments involving *S. oneidensis* and MIL-100, we predicted that viable cells could metabolically regenerate Fe(II) in the framework for continued Cr (VI) reduction and adsorption (Fig. 6a). To determine and compare the recyclability of *S. oneidensis*-reduced materials, samples containing either MIL-100, ferrihydrite, or fumarate were reduced by *S. oneidensis* MR-1 for 24 h before being challenged with ten cycles of $[Cr(VI)]_0 = 0.50$ mM, once every 24 h. For all additions to *S. oneidensis* growing on MIL-100, soluble Cr(VI) was removed below the limit of detection within minutes (Fig. 6b and Supplementary Fig. 12). After ten cycles of Cr(VI) addition, the total amount of Cr(VI) reduced by *S. oneidensis* and MIL-100 per mass of material was $78.7 \pm 0.1$ mg g$^{-1}$ (Supplementary Table 1), which represents a 125-fold increase over MIL-100 alone. When the maximum Cr(VI) capacity was assessed, we found that $320 \pm 1$ mg g$^{-1}$ was reduced after 41 cycles (Supplementary Fig. 13), which compares favorably to other Cr(VI) adsorbents (Supplementary Table 2).

In contrast, biotic and abiotic treatments with fumarate and ferrihydrite showed no appreciable Cr(VI) adsorption and the amount of Cr(VI) in solution increased after each new dose (Fig. 6b). We also investigated whether a microbially reduced soluble iron source could aid in removal of Cr(VI). Indeed, we found that *S. oneidensis* MR-1 grown on Fe(III)-citrate enabled instantaneous reduction after repeated cycles of 0.50 mM Cr(VI) addition (Supplementary Fig. 14). However, similar to our abiotic experiment with supplemented Fe(II), no precipitate was formed

upon Cr(VI) reduction (Supplementary Fig. 11b). These results suggest that, under our pH operating conditions, Cr(III) can only be efficiently removed from solution in the presence of a sorbent material, such as MIL-100.

After Cr(VI) cycling, we characterized MIL-100 using scanning transmission electron microscopy (STEM) and transmission electron microscopy (TEM). Abiotic and biotic samples were morphologically similar (Supplementary Fig. 15). In contrast, elemental mapping revealed that cycled abiotic MIL-100 was primarily composed of Fe with small amounts of Cr while cycled biotic MIL-100 contained substantially more Cr (Fig. 6c, d). This result supports our solution-based measurements and confirms that metabolic reduction of MIL-100 significantly increases its overall capacity for Cr(VI) adsorption. Our cycling data suggests that *S. oneidensis* MR-1 growing on fumarate or ferrihydrite is quickly overwhelmed after repeated challenges with Cr(VI). Indeed, we observed no increase in $OD_{600}$ in the fumarate samples following the first addition of Cr(VI), implying that cell growth was compromised (Supplementary Fig. 16a). In contrast, we measured an increase in Fe(II) over the course of the cycling experiment when *S. oneidensis* was grown on MIL-100. As expected, Fe(II) concentrations decreased immediately after the addition of Cr(VI) but rebounded as viable cells continued to reduce Fe(III) (Supplementary Fig. 16b). We did not measure a similar trend in Fe(III) reduction for *S. oneidensis* MR-1 grown on ferrihydrite that was repeatedly challenged with Cr(VI) (Supplementary Fig. 16b). We also verified the observed trends in Fe(III) reduction during Cr(VI) cycling by directly assaying cell viability. Following the tenth Cr(VI) addition, we plated undiluted aliquots from the MIL-100, ferrihydrite, and fumarate samples onto LB agar. After overnight aerobic growth, the MIL-100 plates exhibited bacterial lawns, indicating viable bacteria

remained after Cr(VI) cycling (Supplementary Fig. 16c, d). In contrast, abiotic samples, biotic ferrihydrite samples, and biotic fumarate samples showed no growth on LB agar plates. Altogether, these results demonstrate that MIL-100 both effectively protects *S. oneidensis* from repeated challenges of cytotoxic Cr(VI) concentrations and synergistically enables Cr(VI) adsorption with rates and capacities that greatly exceed those of biotic metal oxide or abiotic metal-organic frameworks.

## Discussion

We demonstrated that iron-containing metal-organic frameworks can serve as growth substrates for *S. oneidensis*. Biomass measurements and quantitation of Fe(II) levels revealed that these materials behave as respiratory electron sinks similarly to a model iron oxide (namely, ferrihydrite). Across various metrics, biomass yields were comparable for both nontoxic frameworks (MIL-100 and Fe-BTC) and ferrihydrite, despite large differences in the extent of Fe(III) reduction (Fig. 2 and Supplementary Fig. 5). As others have observed similar results for lactate-fed *S. oneidensis* grown on a variety of electron acceptors[17], these results collectively indicate that metal respiration is not growth limiting under certain experimental conditions. While our lactate supplementation/omission and metabolite measurements demonstrated that electron flux primarily derives from lactate catabolism, significant residual Fe(III) reduction was measured in biotic lactate-free experiments compared to abiotic controls (Fig. 3b). Since casamino acids were previously shown to be unimportant for metal oxide reduction[36], this suggests that *S. oneidensis* can metabolize dissolved $H_2$ from the anaerobic chamber atmosphere as an electron donor. Alternatively, residual reducing power (e.g., high NADH or reduced quinone pools) from the pregrowth inoculum could account for lactate-free Fe(III) reduction[37].

Notably, we observed that biomass-normalized Fe(III) reduction rates were accelerated over ferrihydrite with the frameworks MIL-100 and Fe-BTC. Although the preparation conditions of ferrihydrite may affect absolute reduction rates, reduction of the frameworks appeared markedly different from the model iron oxide under our tested conditions. As mentioned, *S. oneidensis*-based reduction may increase solubilization of framework Fe[25]. In contrast to insoluble Fe(III), soluble Fe(III) exhibits faster reduction kinetics by *S. oneidensis*, which could contribute to faster reduction rates of frameworks relative to ferrihydrite. Since others have shown that flavins minimally influence soluble Fe(III) reduction by *S. oneidensis*[24], our *bfe* knockout and flavin supplementation observations further support this mechanism. Nonetheless, substantial levels of Fe(II) were retained within the structures of MIL-100 and Fe-BTC. Moreover, our use of the extracellular cytochrome knockout (Δ*mtrC*Δ*omcA*) demonstrated that canonical iron oxide *S. oneidensis* extracellular electron transfer pathways (namely, MtrCAB) are required for framework reduction. These results, coupled with minimal changes to biomass yields between materials, suggest that differences in Fe(III) reduction rate are directed by material properties instead of changes to metabolism or the cellular route of electron transfer.

Although Fe(III) was present in all tested materials, particle structure and surface area widely differed between the frameworks and ferrihydrite. As has been shown with iron oxides, the extent of Fe(III) reduction by metal-reducing bacteria likely depends on the interplay between particle size, extent of aggregation, and surface area[38–40]. For example, interfacial contact between bacterial outer-membrane cytochromes and iron oxides likely impacts reduction rates, with smaller particles unable to provide adequate contact for reduction[41]. Structurally, MIL-100 and Fe-BTC both possess the same organic ligand, but MIL-100 is more crystalline. While iron oxides with higher degrees of

crystallinity are typically associated with slower Fe(III) reduction rates across the *Shewanella* genus[42,43], we observed that MIL-100 exhibited slightly faster Fe(III) reduction kinetics than the more amorphous Fe-BTC. SEM images showed that MIL-100 and Fe-BTC were highly aggregated, while MIL-88A comprises small, well-defined crystals approximately the same size as *S. oneidensis* MR-1 (Supplementary Fig. 4). Although they are morphologically similar, MIL-100 and Fe-BTC differ in accessible surface area. MIL-100 had the largest Langmuir surface area ($2188 \, m^2 \, g^{-1}$) followed by Fe-BTC ($1512 \, m^2 \, g^{-1}$), ferrihydrite ($311 \, m^2 \, g^{-1}$ (ref. [44]), and MIL-88A ($130 \, m^2 \, g^{-1}$) (Supplementary Table 3). These surface areas closely track our measured Fe(III) reduction rates. Thus, increased surface area likely contributes to significantly faster reduction kinetics by cultures with MIL-100 and Fe-BTC over ferrihydrite. Overall, our results are consistent with previous studies of *S. oneidensis* growth on iron oxides and show that biotic metal-organic framework reduction is governed through a combination of material properties.

Material morphology can also negatively impact Fe(III) reduction and growth, as evidenced by the decline in cell viability for *S. oneidensis* cultures supplemented with MIL-88A. This result was surprising as MIL-88A comprises two known *S. oneidensis* substrates (Fe(III) and fumarate), has a comparable surface area to ferrihydrite, and presumably solubilizes its linker and metal node under biotic conditions (Fig. 4a). Given its sharp, needle-like particle morphology, MIL-88A may compromise cell membranes, resulting in cell lysis[45–47]. Separately, we found that *S. oneidensis* did not exhibit a decline in cell viability in the absence of an electron acceptor and *E. coli* was able to grow unimpeded in the presence of MIL-88A (Supplementary Fig. 3). Unlike *S. oneidensis*, *E. coli* does not readily establish biofilms on redox-active surfaces[48]. While growth dynamics were not continuously monitored, our endpoint measurements revealed that *E. coli* does not exhibit the sharp decline in cell viability that *S. oneidensis* displays in the presence of MIL-88A after 24 h. Taken together, this suggests that the morphology of MIL-88A is cytotoxic to biofilm-forming bacteria, such as *S. oneidensis*, although the exact mechanism for this is unclear.

In one potential application, we showed that *S. oneidensis* operates synergistically with metal-organic frameworks to remediate large quantities of the environmental pollutant, Cr(VI). Specifically, the combination of biologically generated Fe(II) with a high-surface-area porous material, especially MIL-100, enabled rapid reductive adsorption of Cr(VI) from solution. This synergism was supported by low Cr(VI) removal in experiments with material-free biological reduction of soluble Fe(III) and material-free abiotic Fe(II) addition (Supplementary Fig. 11). Although *S. oneidensis* growth on ferrihydrite (a nonporous iron oxide) was capable of generating Fe(II), Cr(VI) adsorption was similarly diminished relative to bioreduced frameworks and cell viability was completely compromised after ten cycles. In addition to exhibiting higher adsorption capacity, MIL-100 was able to shield the bacteria from repeated Cr(VI) challenges. Higher material surface area (Supplementary Table 1) and higher levels of solubilized Fe(II) (Fig. 4c) likely facilitated superior performance by MIL-100 cultures. Specifically, soluble Fe(II) provides a kinetically facile reduction pathway for Cr(VI), while the framework acts as a high-surface area adsorbent. Moreover, Cr(III) can potentially exchange with Fe(III) as a structural node in MIL-100, which may further contribute to the high adsorption capacity measured for this framework[49]. As Fe(III) is continually reduced by *S. oneidensis*, then re-oxidized after Cr(VI) addition, the Cr adsorption capacity of reduced MIL-100 is theoretically only limited by the amount of electron donor in the culture medium. In our long-term cycling experiment, the Cr adsorption capacity for MIL-100 was measured as $320 \, mg \, g^{-1}$, a 523-fold increase relative to abiotic

MIL-100. This capacity corresponds to an amount of reduced Cr (VI) that is approximately 75% of the theoretical maximum based on the abundance of lactate-derived electrons that can be anaerobically metabolized by *S. oneidensis* (four electrons per molecule of lactate)[50]. Based on our metabolite measurements, up to half of the supplemented lactate can be consumed by *S. oneidensis* after 48 h of Fe(III) reduction (Supplementary Fig. 6). Since our long-term cycling experiment occurred well beyond this timescale, the large quantity of Fe(III) reduction and Cr(VI) reduction/adsorption lends further credence to build-up of reduced intracellular metabolites and/or metabolizing of $H_2$ as an electron donor.

Although we examined only a small selection of frameworks, we note that changes to metal node or linker identity can radically alter material structure/functionality and tailor frameworks to specific applications. In addition to iron, metal-organic frameworks can be constructed from a variety of biologically relevant metal nodes, including manganese, chromium, and cobalt[8]. In contrast to metal oxides, the organic linker or other framework components can also be modified to act as secondary electron acceptors, such as with frameworks containing dimethyl sulfoxide[51], porphyrins[52], or conductive linkers[53]. Our Cr(VI) adsorption results with MIL-100 suggest that *S. oneidensis* can be paired with other redox-active frameworks to improve Cr adsorption capacity or remediate additional environmental pollutants, such as U(VI)[54]. Alternatively, metal-organic frameworks may find use as substrates for other electroactive microbes (e.g., *Geobacter* spp.) or as redox mediators that support syntrophic cocultures[55,56]. Overall, our results show that pairing metal-organic frameworks with electroactive microbes can actuate vastly different biological (e.g., Fe(III) reduction rates) and chemical (e.g., Cr(VI) removal) responses, relative to iron oxides. As a result, metal-organic frameworks offer promise for studying the microbe-material interface and optimizing a variety of redox-based technologies.

## Methods

**Synthesis of metal-organic frameworks and iron oxides**. MIL-100 was synthesized by first dissolving $H_3BTC$ (1.68 g, 7.6 mmol) and NaOH (0.91 g, 22.8 mmol) in 24 mL $H_2O$. This solution was added dropwise to $FeCl_2 \cdot 4H_2O$ (2.26 g, 11.4 mmol) dissolved in 97 mL $H_2O$ and stirred for 24 h at room temperature/ambient conditions. The precipitate was isolated, washed with water (3x at 25 °C for 12 h) and ethanol (1x at 25 °C for 12 h), and then dried at room temperature/ambient conditions[57]. Fe-BTC was synthesized by dissolving $H_3BTC$ (0.263 g, 1.2 mmol) and NaOH (0.15 g, 3.8 mmol) in 10 mL of $H_2O$. This solution was added dropwise to $FeCl_3 \cdot 6H_2O$ (0.513 g, 1.9 mmol) dissolved in 10 mL $H_2O$ and stirred for 10 min at room temperature/ambient conditions. The precipitate was isolated, washed with water (3x at 25 °C for 12 h) and ethanol (1x at 25 °C for 12 h), and then dried at room temperature/ambient conditions[58]. MIL-88A was synthesized by stirring a solution of fumaric acid (0.97 g, 8.4 mmol) and $FeCl_3 \cdot 6H_2O$ (2.27 g, 8.4 mmol) in 42 mL of $H_2O$ for 1 h before transferring to a Teflon-lined steel autoclave (Parr). The reactor was heated at 65 °C for 12 h and then cooled down to room temperature. Precipitate from inside the reactor was subsequently washed with ethanol (3x at 25 °C for 12 h) and water (3x at 25 °C for 12 h), then dried at 120 °C for 10 h[59]. Ferrihydrite was synthesized by adding 1 M NaOH dropwise to $FeCl_3 \cdot 6H_2O$ (5.4 g, 20.0 mmol) dissolved in 100 mL $H_2O$ until the pH reached ~7.5. The solid was isolated by centrifugation, washed with water (3x at 25 °C for 10 min), and lyophilized immediately following washing for 48 h[44].

**Strains and culture conditions**. All anaerobic cultures and experiments were performed inside a humidified Coy anaerobic chamber (3% $H_2$/balance $N_2$ atmosphere). *S. oneidensis* MR-1 was obtained from ATCC® (700550™). Mutant strains, $\Delta mtrC\Delta omcA$ and $\Delta bfe$, were generously provided by JA Gralnick (University of Minnesota, Minneapolis, MN). For anaerobic pregrowths, all strains were cultured in SBM buffered with 100 mM HEPES and supplemented with 0.05% casamino acids and 1x Wolfe's mineral mix[60]. The medium was adjusted to a pH of ~7.2. For pregrowth, sodium lactate (20 mM) and sodium fumarate (40 mM) were added as the carbon source and an electron acceptor, respectively. Once the bacteria reached stationary-phase (ca. 18 h), they were pelleted by centrifugation (6000 x g; 20 min) and washed with SBM two times. A final addition of SBM was added to dilute the cells to $OD_{600} = 0.2$.

**Growth of MR-1 on metal-organic frameworks and iron oxides**. Growth of *S. oneidensis* MR-1 on the metal-organic frameworks was measured by CFU counting, nucleic acid staining (Syto™ 9), extracellular protein staining (SYPRO™ Ruby), and total protein biomass (Bradford assay). Cultures containing 20 mM sodium lactate and either MIL-100, Fe-BTC, MIL-88A, or ferrihydrite ($[Fe(III)]_0 = 15$ mM) in fresh SBM were prepared anaerobically and then inoculated with the washed *S. oneidensis* MR-1. CFU/Fe(III) reduction studies and total protein biomass studies used an inoculating $OD_{600} = 0.002$. Nucleic acid staining and extracellular protein staining reduction studies used an inoculating $OD_{600} = 0.02$ to increase the fluorescent signal. Abiotic samples were prepared with the materials in SBM with lactate but with no cells. The cultures were incubated anaerobically at 30 °C. For CFU counting, an aliquot of the culture suspension was removed and serially diluted ($10^0$–$10^{-7}$). Each dilution (100 µL) was plated onto LB agar plates using sterile glass beads and incubated aerobically at 30 °C overnight. Individual colonies on each plate were counted and the initial concentration of *S. oneidensis* MR-1 was calculated. For the staining of bacterial nucleic acids, 200 µL of the culture suspension was removed and pelleted before removing the supernatant. The pellet was resuspended in 100 µL of 0.85% NaCl solution, mixed with 2x Syto™ 9 solution in a 96-well plate, and incubated at room temperature for 15 min in the dark. Fluorescence was measured with an excitation/emission of 485/530 nm using a BMG LabTech CLARIOstar Monochromator Microplate Reader. For the staining of biofilm matrix, 100 µL of the culture suspension was removed and pelleted before removing the supernatant. The pellet was mixed with 200 µL of 1x SYPRO™ Ruby fluorescent dye in a 96-well plate. The plate was incubated in the dark at room temperature for 15 min before measuring fluorescence at an excitation/emission of 460/640 nm using a plate reader. The nucleic acid fluorescence and extracellular protein fluorescence for biotic samples was corrected by subtracting the abiotic total measurement. For the total protein biomass measurements, an 850 µL or 900 µL aliquot of the suspension was pelleted ($10,000 \times g$; 10 min) prior to resuspending in 90 µL of 0.5X SBM buffer. The samples were pelleted once more ($10,000 \times g$; 10 min) and resuspended in 90 µL 0.1 M NaOH. The samples were sonicated in a Bransonic 28100 ultrasonic bath (Branson) for 10 min, then heated at 95 °C for 40 min, and pelleted a final time ($10,000 \times g$; 10 min). The supernatant (10 µL) was mixed with 200 µL of the Better Bradford Reagent (ThermoFisher) in a 96-well plate and analyzed at 595 nm in a plate reader[61]. The measured total protein for biotic samples was corrected by subtracting the average abiotic total protein measurement. The corrected biotic values were used to normalize the Fe (III) reduction rate.

**Reduction of metal-organic frameworks and ferrihydrite**. Reduction of the metal-organic frameworks and ferrihydrite by MR-1 was tested by measuring Fe (II) concentrations in framework-bacteria suspensions over time. Suspensions containing 20 mM sodium lactate and either MIL-100, Fe-BTC, MIL-88A, or ferrihydrite ($[Fe(III)]_0 = 15$ mM) in fresh SBM were prepared anaerobically and then inoculated with the washed *S. oneidensis* MR-1 to $OD_{600} = 0.002$. Abiotic samples were prepared with the materials in SBM with lactate but with no cells. The suspensions were incubated anaerobically at 30 °C. At each time point, the material substrates were gently shaken to fully suspend all materials before removing an aliquot from the culture. Each aliquot was mixed with 6 M HCl in a 1:1 ratio, until full dissolution of the framework or oxide occurred. If the Fe(II) concentrations was large, the sample was further diluted with 1 M HCl. All samples were analyzed for Fe(II) using the ferrozine assay[62]. The ferrozine assay was performed by mixing 15 µL of the diluted sampled with 235 µL of ferrozine solution in a 96-well plate. Absorbance was measured at 562 nm using a microplate reader. Ferrozine solution consisted of ferrozine (45 mg, 87 µmol) and ammonium acetate (22.5 g, 0.29 mol) in 45 mL $H_2O$. Rates of iron reduction were calculated using linear squares regression analysis and were normalized by the sample's corresponding total protein biomass measurement. A similar protocol was followed for reduction of the frameworks by $\Delta mtrC\Delta omcA$ and $\Delta bfe$. For experiments with heat-killed cells, washed *S. oneidensis* MR-1 cells were heated at 80 °C for 10 min. Lysed cell cultures were prepared by sonicating washed *S. oneidensis* MR-1 with a Qsonica 55 probe (Qsonica, LLC, Newtown, CT) for 90 s at 4 °C, maintaining between 10 and 15 W of output energy and alternating between 10 s ON and 5 s OFF.

To assess the amount of Fe(II) contained in biotically reduced MIL-100, Fe-BTC, and MIL-88A, abiotic and framework-bacteria suspensions were prepared identical to those utilized in reduction kinetics experiments. After 48 h, the framework suspension was centrifuged to pellet the solid, and the supernatant was separated. The isolated solid was washed three times with sterile water and resuspended with SBM using the same volume of initially removed supernatant. Fe (II) concentrations were quantified in the whole sample suspension, isolated supernatant and isolated/washed/resuspended framework. As shown in Fig. 4c, the Fe(II) concentrations quantified in isolated supernatant and isolated framework were summed to compare to Fe(II) concentrations measured in the whole sample suspension.

**Cr(VI) removal by reduced frameworks and ferrihydrite**. Cr(VI) removal was tested using metal-organic frameworks or ferrihydrite that had been reduced by *S. oneidensis* MR-1 for 24 h. Cultures containing MIL-100 ($[Fe(III)]_0 = 15$ mM) and 20 mM lactate in SBM were inoculated with washed *S. oneidensis* MR-1 to $OD_{600} = 0.002$ under anaerobic conditions. Abiotic samples were prepared with

MIL-100 in SBM with lactate but with no cells The suspensions were incubated anaerobically at 30 °C for the duration of the experiment. After 24 h, $K_2Cr_2O_7$ was added to each of the cultures ($[Cr(VI)]_0 = 70\,\mu M$) and Fe(II) concentrations were analyzed. Cr(VI) concentration in the supernatant was analyzed using the DPC assay[63]. The DPC assay was performed by mixing 15 µL of sample with 235 µL of DPC solution in a 96-well plate. Absorbance was measured at 540 nm in a microplate reader. DPC solution consisted of DPC (12.5 mg, 103 µmol) in a mixture of 0.5 M $H_2SO_4$ (10 mL) and acetone (10 mL). MIL-100 was tested with higher concentrations of Cr(VI) (0.25, 0.50, and 1 mM) using the same procedure.

**Cr(VI) cycling**. Cr(VI) removal from a supernatant was tested over ten cycles with *S. oneidensis* MR-1 reduced MIL-100, ferrihydrite, and fumarate. MIL-100, ferrihydrite, and fumarate were biotically reduced for 24 h and $K_2Cr_2O_7$ was added to each culture ($[Cr(VI)]_0 = 0.50$ mM). Every 24 h, $K_2Cr_2O_7$ ($[Cr(VI)]_0 = 0.50$ mM) was added, with ten total additions occurring. Fe(II) concentration was monitored using the ferrozine assay, in which aliquots of the suspension were acidified with 6 M HCl in a 1:1 ratio prior to mixing with ferrozine solution and measuring absorbance. Cr(VI) concentration in the supernatant was monitored using the DPC assay. A second cycling experiment was done using the same conditions, but adding 0.5 mM Cr(VI) until no more Cr(VI) was reduced in the supernatant, equaling 41 additions. To determine cell viability after ten additions, aliquots of each sample (100 µL) were plated on LB agar plates 24 h after the last Cr(VI) addition and cells were grown aerobically at 30 °C overnight. Following cycling, cycled samples were washed in fresh $H_2O$ and then dried anaerobically. Prior to STEM imaging, the samples were suspended in ethanol (500 µL) and dropcast on TEM grids. STEM imaging and electron mapping of Fe and Cr was performed using a JOEL 2010F Transmission Electron Microscope.

**Statistical analysis**. Unless otherwise noted, data are reported as mean ± S.D. of $N = 3$ biological replicates, as this sample size was sufficiently large to detect significant differences in means. Significance was calculated using an unpaired two-tailed Student's *t*-test ($\alpha = 0.05$) or analysis of variance (ANOVA) with post-hoc Dunnett's test in Prism 8 by GraphPad Software (San Diego, CA).

**Reporting summary**. Further information on research design is available in the Nature Research Reporting Summary linked to this article.

## Data availability
All data needed to evaluate the conclusions of this paper are present in the paper and/or Supplementary Materials. The source data underlying Figs 2a–d, 3a–c, 4a, c, 5b–f, and 6b and Supplementary Figs S1a–c, S2, S3, S5a–c, S6a–c, S7a, b, S8–S10, S11a, b, S12–S14, and S16a, b are provided as Source Data files. All other data are available from the corresponding author (B.K.K.) by request. The raw data supporting the findings of this study are available from the Texas Data Repository (https://doi.org/10.18738/T8/XEZ6ZY).

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

## Acknowledgements

We thank Hazel Mohamedali, Riley Shuping, Gang Fan, Michael Lucas, Reinaldo Alcalde, and the Werth Lab group for their experimental assistance. S. oneidensis strains Δbfe and ΔmtrCΔomcA were provided by Prof. Jeffrey Gralnick and E. coli MG1655 was provided by Prof. Lydia Contreras. We also thank Prof. Contreras for use of her imaging resources. We acknowledge the use of facilities within the core microscopy lab of the Institute for Cellular and Molecular Biology, University of Texas at Austin and within the X-Ray Diffraction Lab, University of Texas at Austin. S.K.S. was supported through a Provost's Graduate Excellence Fellowship (PGEF). This work was partially supported by the Welch Foundation (Grant F-1929) and the National Science Foundation through the Center for Dynamics and Control of Materials: an NSF Materials Research Science and Engineering Center under Cooperative Agreement DMR-1720595.

## Author contributions

S.K.S., C.M.D., and B.K.K conceived and designed the experiments. S.K.S. and C.M.D. performed the experiments. S.K.S. performed the material synthesis and characterization. B.K.K. supervised the project and generated Fig. 1, Fig. 5a, and Fig. 6a. All authors contributed to data analysis and the writing of the manuscript.

## Competing interests

The authors declare no competing interests.
