## [Peer Review File · Nature Communications]

Reviewers' comments:

Reviewer #1 (Remarks to the Author):

Keitz and co-workers reported the microbial reduction of metal-organic frameworks for the synergistic Cr(VI) removal and characterized the synthesized metal-organic frameworks by SEM, TEM, and Powdered-XRD. Further, authors showed its application that cultures containing *S. oneidensis* and reduced frameworks can remediate lethal concentrations of Cr(VI), Although the manuscript was properly structured for the synergistic Cr(VI) removal by microbial reduction of metal-organic frameworks, but authors used the already known MOFs (Fe-BTC, MIL-100, and MIL-88A) See the reference

i. (a) *Catalysis Today*, 2018, 304, 119-126. (b) *J. Phys. Chem. C*, 2015, 119, 7826-7830.

ii. (a) *Cryst. Growth. Des.*, 2018, 18, 7730-7744. (b) *Dalton Trans*, 2016, 45, 8637-8644.

iii. (a) *RSC ADV*, 2016, 6, 112502-112511. (b) *EurJIC*, 2018, 18, 1909-1915.

in the current work, which does not meet the novelty to publish in "Nature Communication".

The following comments and corrections can be considered to improve the manuscript

1. What is the main interaction mechanism for the uptake of Cr(VI)?
2. It is necessary to compare the adsorption of Cr(VI) capacities with other recent materials.
3. Authors reported that the reduced capacity of Cr(VI) is $320.3 \pm 0.6 \text{ mg g}^{-1}$, is there any possibility to increase the capacity of Cr(VI) by changing metal-organic framework /biological environment.
4. Authors need to provide the necessary characterization of all the synthesized MOFs like FTIR and S-XRD etc.,
5. Authors claimed that the synthesized MOFs are highly stable in the water, what about the pH effect on the MOFs because authors are interested in the biological applications.

Reviewer #2 (Remarks to the Author):

Comments on the paper by Springthorpe et al.

This paper examines the ability of metal-organic frameworks (MOFs) to act as electron acceptors for *Shewanella oneidensis* MR-1, supporting respiratory growth via Fe(III) reduction to Fe(II) via extracellular electron transport (EET). The paper has a number of conclusions:

1. MOFs with iron as the metal, are very good substrates for EET by MR-1
2. Different MOFs give different rates of Fe reduction, depending on several parameters (framework structure, surface area, particle morphology)
3. EET reduction of the MOF requires functional Mtr genes
4. The MOF/bacteria combination is very effective in reducing (and thus remediating) Cr(VI)

The paper is well-written, and the experiments well-explained, but the conclusions overall are compromised by the methodology employed. By this, I mean, there is no good quantification of cell mass during the experiments, so the details of the experiments can't be confirmed. In many cases, in the absence of accurate biomass determinations, it is not at all convincing that the rate numbers are correct. They should be normalized to cell mass, or some determination of cell mass (protein, CHN measurements, etc.). This is particularly true when fairly small differences are seen (the RF excretion mutant strain vs the wild type). A small change in biomass could be responsible for the difference.

This reviewer had a few suggestions to make some of the statements made in the paper a bit more palatable to many of the people in the field.

Line 76: Please describe the formula for SBM – there are a number of basal media for *Shewanella* in the literature, and it takes only a few sentences to make it possible for others to repeat these experiments properly. This becomes important because of some inconsistencies.

Line 77: assuming it is truly a minimal medium, then it should read lactate as THE carbon source.

Line 78: microbiologists will not agree that in a suspension of materials to which *Shewanella* cells can attach and form biofilms (which is shown here), that CFUs are a good way of assessing biomass. What about a protein measurement, or some other indicator.

Line 83: How do you know that it is toxicity? How do you know it is *Shewanella*-specific. Could it just be death due to the inability to respire? Were any controls done with a bit of fumarate to check for toxicity? Were controls done to check the survival of cells in the absence of any electron acceptor.

The same complaints can be made of the fluorescent dyes. They will give a relative idea of when growth is occurring and perhaps, when it stops, but they lack any sort of real quantification. If the biofilm matrix protein increases, does that mean the cells are all in biofilms? What about the planktonic cells. Does each particle, when plated, count as one cell? All sorts of questions arise. There are very good methods for eliminating this problem.

Line 94: The results corroborate the timing of events (and in some cases point out a lack or corroboration), but certainly NOT the cell counts, or any idea of how the cell counts related to biomass. (e.g., are the cells the same size as when the experiment began, are they motile? Do they have the same composition? What is going on?)

Lines 116-120:

There are a lot of assumptions here, and they need to be cleared up before this can be published. There are no good numbers for biomass production, so nothing can be presented as rates/cell Or rates/biomass – as noted below, it appears that growth is done by 12 hours (according to CFU), levels off at 24, and goes down by ~ 40 hr.

Experiments done in stationary phase are prone to many artifacts and details that are not understood.

Second, there is no way to tell whether cells are dead, alive, or just non-growing. Also, if the water is being sampled, (or the cells are being bound strongly to one particle or another, then CFUs will be low, and get lower with time, if biofilms are being formed (each MOF may count as 1 colony, even if it has a biofilm on it with many cells).

To this end, Figure 2 is confusing: (a) is said to be CFU counts and Fe(III) reduction, but (d) is Fe(III) reduction!

All four are shown in (a) and (d), but only MIL-100 & Ferrihydrite (FH) are shown in b & c.

There is inconsistency between the figures:

The cell counts go up from 10^6 to $5-9 \times 10^7$ or about 50X -- -- about 5-6 generations in 12 hrs.

Then no more – gen. time about 2 hours

NA acids: increase about 5X in 10 hours, 10 X in 24 hours (for ferrihydrite)

Biofilm matrix: about 2X in 10 hours – about 4-5 X in 24 hours (same for FH)

Fe(II): MIL-100 about 2X in 12 hr; 12 X in 24 hr

FE-BTC about 2X in 12 hr; about 6X in 24 hr

FH about 2X in 12 hr; about 4X in 24 hr

MIL88A – about the same as Ferrihydrite

How much reduced iron is bound to the matrices and/or Ferrihydrite?

Where are the cells?

Any biofilms seen? – looks like the same amount for the three positives

In no case was the amount of lactate monitored – why not? If the lactate was used up, reduction would stop, but one can't know from these experiments.

As noted above, Figure 3 has rates that are measured after 96 hours:

Why 96 hours? Everything was done by 24 hours

Rate for MIL-100 at 24 hr would be about .06 (as seen in Fig. 2), so it went down !!

All sorts of things can happen in stationary phase!

Need to normalize the two positive numbers to the amount of biomass!

Likely that the RF excretion mutant grows more slowly

Really would like to have rates/time/cell !!

In (d), the rate of the RF minus reduction was only .038 (smaller than the Bfe mutant in (c) --

Why was 10X RF added? This is an amount probably never seen in nature!

This is a big controversy in the literature that is not discussed here

See Okamoto papers, proposing that this is a lab artifact, and that the

Role of RF is to bind to the Mtr proteins!

This should at least be discussed

Also, it would be nice to have similar data for MIL-88 and F-BTC too

On line 141, where it summarizes the impact of particle properties on reduction rates, it might be useful to cite an old paper (1980s) of Burdige and others (also done with MR-1 I believe) on reduction of Mn oxides, that had almost the same conclusions – crystallinity, surface area, and degree of hydration of the minerals controlled the rates of metal reduction. This would be fascinating to do in conjunction with MOFs!.

On line 160, it would be appropriate to say that the MtrCAB pathway is required for MOF reduction. It is not clear at all from this work what controls the rate, as noted in the paper, -- it could be the number of MtrCAB sites/cell, but it could also be the nature of the MOFs, as proposed above.

On line 163, it would be proper to say that the Mtr proteins enable EET to both some soluble electron acceptors (metals and non-metals) as well as insoluble metal oxides and hydroxides. There is ample discussion of this in many different papers (soluble U(VI) and Cr(VI) (which become insoluble and lethal if reduced inside the cell, for instance)).

Line 171: Unless other flavins were used, this should read: Exogenous riboflavin RF improves reduction ...

It is noted again here, that 10 uM makes it work in the lab, while nM levels (usually the level found in cultures and in nature) have only minimal impact. The 10 uM levels almost certainly act as electron shuttles, but so do many other quinone-type compounds. Were these experiments done with MIL-88A, or with Ferrihydrite?

Line 177: As noted above, were accurate biomass numbers (that would allow normalization to rates/biomass) obtained in any of these experiments?

Line 179: should say ... compared to wild-type MR-1.

Line 188. I am not aware of MR-1 being capable of reduction of heavy metals, but will look in reference 30. The authors should check this. I assume by this they mean Hg or Pb. Cr is a transition metal and is very easily reduced by MR-1.

The experiments with the Cr(VI) are very interesting and should be the crux of the paper.

In summary, this paper needs some careful microbiology (microbial physiology) to back up the statements made with quantitative data. It is definitely worth the effort, as the basic findings are interesting, and probably correct. If so, it is a nice step forward for both the basic science, and many future applications.

One final point, in the figure showing the reduction of iron biotically and abiotically without a carbon source, it is noted that the MOF contains oxalate, and that this might be used as a carbon source. To my knowledge, *Shewanella* doesn't use oxalate, and also oxalate is often used as a solubilizing agent for oxidized iron and/or manganese.

I hope this paper is improved and published
Ken Nealson

Reviewer #3 (Remarks to the Author):

Springthorpe et al.

The authors show that the Fe(III)-reducing bacterium *Shewanella oneidensis* MR-1 can use ferrous iron containing metal-organic frameworks as electron acceptors for growth. Three different types (MIL-100, Fe-BTC and MIL-88A) of MOF were used and compared to amorphous ferrihydrite. Moreover, in a practical application the authors found rapid abiotic reduction of Cr(VI) by Fe(II) in the metal-organic framework and Fe(II) released from the framework. The rapid reduction kinetics observed reduces Cr(VI) toxicity to *S. oneidensis*. The findings are discussed in a wider context of the usefulness of metal-organic frameworks.

General comments

While the experiments were carefully conducted, the data are mainly descriptive and do not provide a thorough mechanistic explanation for why some metal-organic frameworks are better in supporting growth and Cr(III) sorption than others. The observation that *S. oneidensis* can use Fe-containing metal-organic frameworks as electron acceptor by itself is not surprising.

The most interesting aspect of the paper is that reduced MIL-100 has the interesting capacity to reduce Cr(VI) and sorb presumably Cr(III) into its framework. And this is different from the other materials tested. This begs the question for a molecular mechanistic explanation.

There is also a paucity in testing and deducing the individual roles of framework structure, surface area, and particle morphology. The data presented measure these parameters for the three materials tested; however, contrary to the 'advertisement' in the abstract, the paper does not present a systematic deconvolution and contribution of these individual elements. Along these lines, the proposed advantages of metal-organics for a highly tunable structure-function relationship was not tested.

The manuscript would have been strengthened, if other Fe(III) respiring microorganisms would have been tested as well.

For a larger audience, the three metal-organic frameworks need to be introduced better including the important differences.

Growth experiments

I am still not sure that I understand exactly the growth and metabolite experiments. First of all, the measurement of cell density (cfu) is sufficient to demonstrate growth, and DNA and biofilm protein/EPS staining is unnecessary. In fact, it is unclear why biofilm protein (which protein?) staining was conducted, and how these molecular fluorescence measurements were integrated. What additional aspect do these data show?

The microorganism is grown in anaerobic medium containing lactate (20mM) as catabolic electron donor. In contrast to the calculations by the authors (Line 242 and further), *S. oneidensis* oxidizes lactate incompletely to acetate when growing anaerobically. This needs to be considered. Based on Fig. 2, growth ceases after 12 hours, yet most of Fe(II) is released (for MIL-100) after that time point. This finding remains unexplained. It is also difficult to understand how the reduction of 0.5 mM Fe(III) is sufficient to support growth to a cell density of 10⁸ cells. Are we looking here at a dissolution kinetics?

Biological rates should be given as specific rates and normalized to the biomass (Fig. 3). Also based on the data of Fig. 3c, the involvement of flavins is slight, and the statement in the following sentence that '...flavins are important contributors to Fe(III) reduction...' is not warranted (176-181).

It is argued (Fig. S7 and 152-158) that *S. oneidensis* can reduce the Fe(III)-organic framework, which leads to a significant but not complete dissolution of the framework complex. How then, can the framework stability be maintained if Fe(II) partially dissolves? How is that 'stability' determined?

Cr(VI) reduction experiments

These experiments are very interesting! The experiments were conducted with 24 hrs old cultures. What was the state of metabolic activity in these cultures? Was there any lactate left at 24hrs or Fe(III) reduction activity left? Was the Fe(III) reduction rate observed due to microbial activity or the kinetics of Fe(II) dissolution (203)?

What mechanism do the authors propose to explain why no corresponding decrease in Cr(VI) was observed, despite appreciable concentration of Fe(II) in the ferrihydrite experiment (Fig.4)? Please discuss in text.

Discussion

The discussion is lengthy and redundant to a large part. Again here, the emphasis is on metal-organic frameworks in general rather than on the issues identified in this study. The 'advantage of these materials is their high degree of synthetic tunability' was emphasized, but little systematic, mechanistic studies for a better fundamental understanding how these diverse structures explain different microbial reduction rates was provided. Furthermore, significant space (282-311) was devoted to elaborate on the potential of microbial-metal-organic framework systems, including the limitations on stability in water. This is interesting but unrelated to the data presented. The authors should have presented experimental data that address these points. In the absence of that, these sections read more like an advertisement for metal-organic frameworks rather than a critical discussion of the data presented.

The mode of toxicity of MIL-88A should be discussed.

Some editorial comments

In general, the microorganism should be addressed as '*S. oneidensis* MR1' rather than 'MR1' in the text.

86 Which biofilm matrix protein is stained by SYPRO Ruby?

95 What is 'some' (emphasis added) iron-based metal-organic framework?

147 The acronym 'PXRD' needs to be explained.

288 What is 'electroactive' physiology?

Editor/Reviewers' comments (black) and authors' response (blue):

Reviewer #1

We thank the reviewer for their input on the manuscript and have addressed individual reviewer comments below. While these MOFs as well as Cr(VI) adsorption have been reported by others, we believe the novelty of our study derives from understanding and leveraging the bacteria-MOF interactions: (1) it is the first demonstration that metal-organic frameworks can serve as growth/reduction substrates for electroactive bacteria and (2) the MOF-bacteria pairing exhibits dramatic improvement in Cr(VI) removal over abiotic MOFs, isolated bacteria, and bacteria-iron oxide suspensions. Notably, we show through a suite of microbiological and material characterizations that frameworks enable substantially faster Fe(III) reduction rates, relative to iron oxides. Moreover, we find these materials remain structurally stable and possess Fe(II) within the framework. The high levels of Fe(II) and surface area from MOFs appear critical for facilitating superior Cr adsorption capacity.

1. What is the main interaction mechanism for the uptake of Cr(VI)?

We believe the main interaction mechanism for uptake of Cr(VI) is the redox reaction between biologically-generated Fe(II), both insoluble and soluble, and Cr(VI). At the pH of our reaction medium (pH 7.2), Cr(III) remains soluble and can be adsorbed to the MOF. This is supported by our data from Figures 5 and S10, which shows that Cr(VI) is reduced quickly and total dissolved Cr is removed from the supernatant more slowly and adsorbed to the MOF. The high surface

area of the frameworks likely facilitates superior Cr(VI) uptake relative to ferrihydrite (see Table S3 and Figure 5). To improve understanding of the bio- and material-based means of Cr(VI) uptake, we have rewritten our results (see Lines 250-267) and discussion sections (see Lines 368-391) and highlighted our experiments that disentangle the different mechanisms. To further improve this understanding, we have also added a new figure (Figure 4) that contains previous SI figures discussing frameworks stability and Fe(II) within the framework.

2. It is necessary to compare the adsorption of Cr(VI) capacities with other recent materials.

We thank the reviewer for bringing this to our attention. In Table S2, we provided reported Cr(VI) capacities for both biotic and abiotic agents.

3. Authors reported that the reduced capacity of Cr(VI) is 320.3 ± 0.6 mg g⁻¹, is there any possibility to increase the capacity of Cr(VI) by changing metal-organic framework /biological environment.

We do believe that this is possible. By using different redox-active MOFs that either have improved adsorption capabilities over MIL-100 or can be reduced at faster rates, the capacity could likely be improved. Please see the last paragraph of the discussion.

4. Authors need to provide the necessary characterization of all the synthesized MOFs like FTIR and S-XRD etc.,

We appreciate the reviewer's suggestion concerning FTIR and SXR, but we also believe that this would be unnecessary in this case. These MOFs have already been fully characterized by others and are well-known in the literature. We feel that PXRD and surface area measurements are sufficient to confirm the identity of the MOFs and their stability pre/post-reduction.

5. Authors claimed that the synthesized MOFs are highly stable in the water, what about the pH effect on the MOFs because authors are interested in the biological applications.

We thank the reviewer for noting the influence of pH. In our experiments, we did analyze the stability of the MOFs under abiotic culture conditions (pH=7.2) after exposure for 48 h, the typical timescale of our experiments. We found no significant Fe(III) leaching or change in PXRD after exposure to pH=7.2 media, as discussed in lines 76-89, which suggests minimal degradation. Because we were not using pH conditions aside from that of the culture medium, we believe it is unnecessary in our study to examine the stability of the MOFs at a variety of pH conditions.

Reviewer #2 (Remarks to the Author):

Line 76: Please describe the formula for SBM – there are a number of basal media for *Shewanella* in the literature, and it takes only a few sentences to make it possible for others to repeat these experiments properly. This becomes important because of some inconsistencies.

We apologize for not being clearer with our formulation of SBM. While the exact formulation is stated in the materials and methods, we have added mention in the results section (Lines 81-83) of the primary buffering agent (100 mM HEPES) and the presence of casamino acids (0.05%) and trace minerals (1X Wolfe's Mineral Solution).

Line 77: assuming it is truly a minimal medium, then it should read lactate as THE carbon source.

We have changed the manuscript to state lactate is "the carbon source" (see Line 95).

Line 78: microbiologists will not agree that in a suspension of materials to which *Shewanella* cells can attach and form biofilms (which is shown here), that CFUs are a good way of assessing biomass. What about a protein measurement, or some other indicator.

We thank the reviewer for their analysis of the growth and the shortcomings of the methods used. Given the dichotomy of our MIL-88A and ferrihydrite/MIL-100/Fe-BTC CFU counts and the similarity in counts between positive-growth materials, it appears that CFUs can reveal some information with respect to how each material affects cell growth/viability. However, we acknowledge the inadequacies of CFUs and that they likely violate of the assumption of 1 colony=1 bacterium in the presence of materials. Thus, we have measured total protein content using the Bradford

assay, as an additional metric to assess biomass accumulation (see Figure 2). We have also repeated all of our knockout strain (bfe and mtrComcA) and flavin supplementation experiments, but with the inclusion of total protein measurements in order to normalize all reported rates by mg of protein (see Figure 3, Figure S6, Figure S9). We note that Bradford can exhibit preference for certain polypeptide sequences, but our attempts to utilize the more promiscuous BCA assay proved untenable due to high background signal in the presence of high Fe(II) and porous materials. Nonetheless, Bradford measurements appear to provide reasonable results that corroborate our CFU and fluorescent dye observations.

Line 83: How do you know that it is toxicity? How do you know it is Shewanella-specific. Could it just be death due to the inability to respire? Were any controls done with a bit of fumarate to check for toxicity? Were controls done to check the survival of cells in the absence of any electron acceptor.

We thank the reviewer for their concern and have improved clarity of this in our results (see Lines 100-110) and discussion (see Lines 358-367). We analyzed CFU counts of *S. oneidensis* in the absence of an electron acceptor over the same timescale as material-suspension experiments (see Figure S3) and observed relatively constant CFUs. This would suggest that *S. oneidensis* does not exhibit a decrease in viability due to lack of respiration. We also examined the anaerobic growth of *Escherichia coli* MG1655 in culture with MIL-88A+no electron acceptor and MIL-88A+40 mM fumarate (see Figure S3). In both cases, CFU counts increased over 24 h to relatively comparable extents. This result suggests that the effect of MIL-88A isn't exhibiting a general bactericidal effect. Thus, we speculate that MIL-88A's unique particle morphology (relative to the other tested materials) coupled with differences in biofilm formation between *E. coli* and *S. oneidensis* might account for differential effects on cell viability. Ongoing work in our lab seeks to better understand the mechanistic basis for these differences (e.g., testing different MIL-88A particle morphologies), but we believe these experiments are best suited for a separate manuscript.

The same complaints can be made of the fluorescent dyes. They will give a relative idea of when growth is occurring and perhaps, when it stops, but they lack any sort of real quantification. If the biofilm matrix protein increases, does that mean the cells are all in biofilms? What about the planktonic cells. Does each particle, when plated, count as one cell? All sorts of questions arise. There are very good methods for eliminating this problem.

We address these questions below.

Line 94: The results corroborate the timing of events (and in some cases point out a lack or corroboration), but certainly NOT the cell counts, or any idea of how the cell counts related to biomass. (e.g., are the cells the same size as when the experiment began, are they motile? Do they have the same composition? What is going on?)

We agree with the reviewer that cell counts may not necessarily correlate with biomass or other changes in cell composition (see Kostka et al. *Appl. Environ. Microbiol.* 2002). Although we did not directly measure changes in cell size, total protein measurements should capture changes in cellular size/composition and allowed us to normalize Fe(III) reduction rates between different materials. Although cell density may have plateaued, continued respiration is likely required for active metabolism/biofilm maintenance (Saville et al. *J. Bacteriol.* 2011, Rowe et al. *mBio* 2018).

Lines 116-120: There are a lot of assumptions here, and they need to be cleared up before this can be published. There are no good numbers for biomass production, so nothing can be presented as rates/cell or rates/biomass – as noted below, it appears that growth is done by 12 hours (according to CFU), levels off at 24, and goes down by ~ 40 hr.

Experiments done in stationary phase are prone to many artifacts and details that are not understood.

As addressed above, we have normalized all Fe(III) reduction rates to Bradford-measured total protein content at the end of each kinetic run. The long timescale kinetics (48 h or greater) were chosen as these results inform our Cr(VI) removal that occur on these timescales. Moreover, detecting biomass/Fe(II) levels at early timepoints was near our limits of detection and made distinguishing trends between materials challenging. Based on CFU counts and total protein measurements, biomass levels are relatively invariant between *S. oneidensis* cultures when grow on MIL-100, Fe-BTC, or ferrihydrite. This held true for these three materials when biomass was measured at 48 h (Figure S6) and 120 h (Figure 2). Thus, although cells have likely entered stationary-phase, the large differences in kinetics suggest material-driven effects on Fe(III) reduction and that reduction is not tied to further biomass accumulation. Similar results have been observed by Kostka et al. *Appl. Environ. Microbiol.* 2002 on >48 h timescales and we have further elaborated on this in the discussion section (Lines 314-339). As mentioned above, respiration is also likely required for metabolism/biofilm maintenance.

Second, there is no way to tell whether cells are dead, alive, or just non-growing. Also, if the water is being sampled, (or the cells are being bound strongly to one particle or another, then CFUs will be low, and get lower with time, if biofilms are being formed (each MOF may count as 1 colony, even if it has a biofilm on it with many cells).

We thank the reviewer for their comment and agree that this is one of the many limitations of using CFU counts for quantifying growth. To be clear, we plated aliquots of material suspensions, and are ostensibly collecting biofilm and planktonic cells. It is noteworthy that in the absence of material toxicity, MIL-100, Fe-BTC and ferrihydrite all yielded comparable stationary phase cell densities, even if their underlying particle characteristics were different. Moreover, given the likely violation of the 1 CFU=1 bacterium assumption, CFU counts likely underestimate cell concentrations in our experiments. With this in mind, we still see concomitant increases in CFU counts and Fe(II), strongly suggesting active metabolism by *S. oneidensis*. Our new experiments also measured concomitant increases in total protein content (as measured by Bradford) and Fe(II), further supporting this conclusion (see Figure 2). In a revision experiment, we further probed the influence of active metabolism by testing sonically lysed and heat-killed cells with MIL-100 and Fe-BTC suspensions (see Figure 3a). In agreement with the biomass measurements, neither of these metabolically compromised cell-material suspensions exhibited increases in Fe(II) levels, relative to those with viable whole-cell inocula. Furthermore, we performed concomitant Fe(II) and metabolite quantification for *S. oneidensis* growing with MIL-100, Fe-BTC, and ferrihydrite (Figure S7). These measurements reveal general trends of Fe(II)/pyruvate/acetate generation that follow lactate consumption, further supporting active cellular metabolism on frameworks akin to ferrihydrite.

To this end, Figure 2 is confusing: (a) is said to be CFU counts and Fe(III) reduction, but (d) is Fe(III) reduction!

We apologize for the confusion. The caption for Figure 2 (a) mistakenly referred to Fe(III) reduction, when it should have just referred to CFU counts. We have fixed this in revisions and moved the concomitant Fe(III) reduction data for this figure to the SI (see Figure S2).

All four are shown in (a) and (d), but only MIL-100 & Ferrihydrite (FH) are shown in b & c.

We thank the reviewer for pointing out this confusion. As our new total protein measurements are more informative than the fluorescent dye experiments, we performed them with all four materials and included them in the main text results (see Figure 2). Since the signal-to-noise can be problematic for the dye experiments, we only chose the two extremes in positive growth and differential reduction (namely MIL-100 and ferrihydrite) to further corroborate the CFU counts and total protein measurements. However, to address your comments and the concerns of Reviewer 3, who felt they were not necessary to show, we have moved these two figures to the SI (see Figure S5).

There is inconsistency between the figures:

The cell counts go up from 10^6 to $5-9 \times 10^7$ or about 50X -- -- about 5-6 generations in 12 hrs.

Then no more – gen. time about 2 hours

NA acids: increase about 5X in 10 hours, 10 X in 24 hours (for ferrihydrite)

Biofilm matrix: about 2X in 10 hours – about 4-5 X in 24 hours (same for FH)

Fe(II): MIL-100 about 2X in 12 hr; 12 X in 24 hr

FE-BTC about 2X in 12 hr; about 6X in 24 hr

FH about 2X in 12 hr; about 4X in 24 hr

MIL88A – about the same as Ferrihydrite

We apologize for the confusion. The initial cell densities for the fluorescent staining experiments were increased relative to the CFU counting experiments by an order of magnitude, as to aid in fluorescence quantification. For the CFU counting experiment, initial OD600 was 0.002, while for the staining experiments, initial OD600 was 0.02. Assuming cell counts plateau at the same level, it is expected that relative changes in NA acids/Biofilm matrix signal would at most be about an order of magnitude. As seen in the CFU counts and total protein measurements and as noted by the reviewer, growth seems to only occur in the first 12 h. Thus, correlations between cell counts and measures of biomass in that time frame should be compared. Beyond that time, iron reduction seems decoupled from cell growth and again is likely more tied to metabolic/biofilm maintenance.

How much reduced iron is bound to the matrices and/or Ferrihydrite?

Please see revised Figure 4c. We've also addressed this topic in the results (lines 204-207).

Where are the cells? Any biofilms seen? – looks like the same amount for the three positives

We did not directly observe any biofilms. However, a number of our experiments suggest biofilm formation is occurring. First, SYPRO Ruby measures biofilm matrix proteins and increases in fluorescence intensity over time for both MIL-100 and ferrihydrite, suggesting biofilm formation. Second, MIL-88A shows *Shewanella*-specific toxicity (but not for *E. coli*), which suggests that biofilm formation or direct cell-material contact is a prerequisite for Fe(III) reduction. Finally, we could not measure an OD600 signal in post-reduction supernatant samples, suggesting an insignificant population of planktonic cells. Our results are consistent with previous studies on metal oxides and we feel that direct visualization of biofilm structure/dynamics is best left to a future manuscript.

In no case was the amount of lactate monitored – why not? If the lactate was used up, reduction would stop, but one can't know from these experiments.

We thank the reviewer for their suggestion to monitor lactate. As mentioned, we performed metabolite and Fe(II) quantification after 48 h for MIL-100, Fe-BTC, and ferrihydrite (Figure S7). Indeed, after 48 h, up to ca. 50% of the supplemented lactate can be utilized. Although less than the expected stoichiometric amount of Fe(II) was generated at this timescale (likely due to accumulation of metabolic intermediate and/or intracellular reductants), our long-term Cr(VI) cycling experiment suggest that other electron donors (e.g., anaerobic chamber H₂) can be utilized by the cells. We have added these results to the revised manuscript and included discussion on Lines 314-327.

As noted above, Figure 3 has rates that are measured after 96 hours:

Why 96 hours? Everything was done by 24 hours

While the reviewer notes that growth was assessed for 24 h, we found that iron reduction continued on much longer timescales than this. Even after cell growth ceases at 12 h, it is likely that respiration for cell maintenance is required (Saville et al. *J. Bacteriol.* 2011, Rowe et al. *mBio* 2018) and iron reduction kinetics appears constant and linear. Additionally, biomass and Fe(II) levels were within more detectable ranges with less variation on >12 h timescales, and these timescales are more informative for Cr adsorption.

Rate for MIL-100 at 24 hr would be about .06 (as seen in Fig. 2), so it went down !!

All sorts of things can happen in stationary phase!

We thank the reviewer for noting these points. As mentioned before, linear reduction rates are observed past ~12 h for each non-toxic material despite minimal change in CFUs or total protein content. We believe the reviewer's calculated rate from Figure 2 is incorrect as it combines single endpoints from earlier phases of reduction/growth, where biomass accumulation may be more directly tied to reduction rate. In contrast, our reduction rates were calculated over the more time resolved and longer period that exhibits linear Fe(III) reduction and an apparent decoupling from cell growth. Thus, our results support the conclusion that, similar to ferrihydrite, MIL-100 and Fe-BTC do support cell growth relative to the electron acceptor free control (Figure S3), but we do not speculate on how the materials affect growth rate/reduction rate at earlier timepoints.

Need to normalize the two positive numbers to the amount of biomass!

Likely that the RF excretion mutant grows more slowly

Really would like to have rates/time/cell !!

We thank the reviewer for raising these points. We have repeated these kinetics measurements and included endpoint total protein measurement to normalize rates. Indeed, the RF excretion mutant had lower biomass, and the normalized rate was similar to that of MR-1. We have included mention of this in the revised manuscript (see Line 180-183).

In (d), the rate of the RF minus reduction was only .038 (smaller than the

Bfe mutant in (c) --

Why was 10X RF added? This is an amount probably never seen in nature!

This is a big controversy in the literature that is not discussed here

See Okamoto papers, proposing that this is a lab artifact, and that the

Role of RF is to bind to the Mtr proteins!

This should at least be discussed

Also, it would be nice to have similar data for MIL-88 and F-BTC too

We agree with the reviewer that the specific role of flavins in our experiments is not precisely clear. Upon the reviewer's suggestion, we repeated all flavin supplementation experiments and normalized by endpoint biomass and included a more physiologically relevant (but still on the high end) concentration of riboflavin (1 μM , see Figure S9). We also repeated this experiment for Fe-BTC. In all cases, we observed no significant increase in biomass normalized Fe(III) reduction rates. As a result, we moved this figure to the SI. We also added discussion (see Lines 176-184 and Lines 332-333) on the potential role of flavins in our system.

On line 141, where it summarizes the impact of particle properties on reduction rates, it might be useful to cite an old paper (1980s) of Burdige and others (also done with MR-1 I believe) on reduction of Mn oxides, that had almost the same conclusions – crystallinity, surface area, and degree of hydration of the minerals controlled the rates of metal reduction. This would be fascinating to do in conjunction with MOFs!

We thank the reviewer for this reference suggestion and have incorporated it into our revised manuscript along with some discussion on the roles of particle morphology and surface area (see Lines 358-367).

On line 160, it would be appropriate to say that the MtrCAB pathway is required for MOF reduction. It is not clear at all from this work what controls the rate, as noted in the paper, -- it could be the number of MtrCAB sites/cell, but it could also be the nature of the MOFs, as proposed above.

We apologize for overstating this and have revised the language in the manuscript to say that MOF reduction is primarily mediated by extracellular cytochromes (line 187-188).

On line 163, it would be proper to say that the Mtr proteins enable EET to both some soluble electron acceptors (metals and non-metals) as well as insoluble metal oxides and hydroxides. There is ample discussion of this in many different papers (soluble U(VI) and Cr(VI) (which become insoluble and lethal if reduced inside the cell, for instance)).

We thank the reviewer for bringing this to our attention and have included it in the Cr(VI) results as well as the discussion (see Line 252 and Line 400).

Line 171: Unless other flavins were used, this should read: Exogenous riboflavin RF improves reduction ... It is noted again here, that 10 μM makes it work in the lab, while nM levels (usually the level found in cultures and in nature) have only minimal impact. The 10 μM levels almost certainly act as electron shuttles, but so do many other quinone-type compounds. Were these experiments done with MIL-88A, or with Ferrihydrite?

We addressed this point above. We did not examine MIL-88A since it is toxic to *S. oneidensis*. Previous studies have examined the role of flavins in ferrihydrite reduction (see ref. 22)

Line 177: As noted above, were accurate biomass numbers (that would allow normalization to rates/biomass) obtained in any of these experiments?

Addressed above.

Line 179: should say ... compared to wild-type MR-1.

We thank the reviewer for bringing this to our attention and have corrected this.

Line 188. I am not aware of MR-1 being capable of reduction of heavy metals, but will look in reference 30. The authors should check this. I assume by this they mean Hg or Pb. Cr is a transition metal and is very easily reduced by MR-1.

We thank the reviewer for bringing this to our attention and have switched "heavy metal" to "transition metal" to improve clarity.

The experiments with the Cr(VI) are very interesting and should be the crux of the paper.

In summary, this paper needs some careful microbiology (microbial physiology) to back up the statements made with quantitative data. It is definitely worth the effort, as the basic findings are interesting, and probably correct. If so, it is a nice step forward for both the basic science, and many future applications.

One final point, in the figure showing the reduction of iron biotically and abiotically without a carbon source, it is noted that the MOF contains oxalate, and that this might be used as a carbon source. To my knowledge, *Shewanella* doesn't use oxalate, and also oxalate is often used as a solubilizing agent for oxidized iron and/or manganese.

We removed the discussion on the MOF containing oxalate.

Reviewer #3 (Remarks to the Author): Springthorpe et al.

The authors show that the Fe(III)-reducing bacterium *Shewanella oneidensis* MR-1 can use ferrous iron containing metal-organic frameworks as electron acceptors for growth. Three different types (MIL-100, Fe-BTC and MIL-88A) of MOF were used and compared to amorphous ferrihydrite. Moreover, in a practical application the authors found rapid abiotic reduction of Cr(VI) by Fe(II) in the metal-organic framework and Fe(II) released from the framework. The rapid reduction kinetics observed reduces Cr(VI) toxicity to *S. oneidensis*. The findings are discussed in a wider context of the usefulness of metal-organic frameworks.

General comments

While the experiments were carefully conducted, the data are mainly descriptive and do not provide a thorough mechanistic explanation for why some metal-organic frameworks are better in supporting growth and Cr(III) sorption than others. The observation that *S. oneidensis* can use Fe-containing metal-organic frameworks as electron acceptor by itself is not surprising.

We reorganized the results and have added a more thorough mechanistic discussion on why frameworks show differences in Fe(III) reduction and Cr(VI) reduction/sorption compared to ferrihydrite (see Lines 250-267 and Lines 368-391). We disagree that *S. oneidensis* respiration on Fe-MOFs is not surprising. Our results with MIL-88A suggest that just because a material contains Fe(III), does not mean it will be a good substrate for electroactive microbial growth.

The most interesting aspect of the paper is that reduced MIL-100 has the interesting capacity to reduce Cr(VI) and sorb presumably Cr(III) into its framework. And this is different from the other materials tested. This begs the question for a molecular mechanistic explanation.

We have added a more detailed mechanistic explanation (see Lines 250-267 and Lines 368-391).

There is also a paucity in testing and deducing the individual roles of framework structure, surface area, and particle morphology. The data presented measure these parameters for the three materials tested; however, contrary to the 'advertisement' in the abstract, the paper does not present a systematic deconvolution and contribution of these individual elements. Along these lines, the proposed advantages of metal-organics for a highly tunable structure-function relationship was not tested.

We agree with the reviewer that our results do not systematically deconvolute these factors and have changed the abstract/intro to emphasize that MOFs support bacterial growth and how this influences Cr(VI) removal. The effects of surface area, particle morphology, and particle size are still controversial when metal oxides are used as electron acceptors (see Discussion lines 340-367). Our results suggest that surface area is one contributor to the larger Fe(III) reduction rates observed with the MOFs, but we believe a more systematic study of material effect is better left to a future manuscript. Our results show that MOFs have advantages over metal oxides, laying the foundation for a more thorough exploration of their tunable structures and function in the future.

The manuscript would have been strengthened, if other Fe(III) respiring microorganisms would have been tested as well.

While we agree with the reviewer that testing other Fe(III) microorganisms with these materials would be fascinating, we believe that this would be outside the scope of this study.

For a larger audience, the three metal-organic frameworks need to be introduced better including the important differences.

We thank the reviewer for their suggestion to improve interest to a larger audience and have revised the manuscript to better introduce the materials used and their key differences (see Lines 77-79).

Growth experiments

I am still not sure that I understand exactly the growth and metabolite experiments. First of all, the measurement of cell density (cfu) is sufficient to demonstrate growth, and DNA and biofilm protein/EPS staining is unnecessary. In fact, it is unclear why biofilm protein (which protein?) staining was conducted, and how these molecular fluorescence measurements were integrated. What additional aspect do these data show?

We thank the reviewer for their comments regarding the growth data. We included the nucleic acid and biofilm stain, which stains most classes of proteins (glycoproteins, phosphoproteins, lipoproteins, calcium binding proteins, fibrillar proteins), data to further support that the cells were growing on our substrates. While they are an effective means of analyzing cell counts, CFUs are most effective for soluble substrates. As noted by Reviewer 2, when an insoluble substrate is used, the typical assumption of 1 colony=1 bacterium may be violated. Given its limitations with insoluble material, we decided that these fluorescent stains could lend more evidence to our assertions that *S. oneidensis* was growing. However, in light of Reviewer 2's comments, we believe that using a different measure of biomass (namely total protein content) is a better indicator and have moved the SYPRO Ruby and Syto9 data to the SI. We repeated all reduction rate experiments and included the measure of endpoint total protein content to normalize our calculated Fe(III) reduction rates.

The microorganism is grown in anaerobic medium containing lactate (20mM) as catabolic electron donor. In contrast to the calculations by the authors (Line 242 and further), *S. oneidensis* oxidizes lactate incompletely to acetate when growing anaerobically. This needs to be considered.

We apologize for the confusion. We have accounted for the incomplete oxidation of lactate to acetate in our calculations for electron stoichiometry (at most, 4 electrons per molecule lactate) and theoretical Cr(VI) capacity based on lactate consumption. Per Reviewer 2's suggestion, we measured lactate, pyruvate, and acetate levels during *S. oneidensis* growth on the MOFs and ferrihydrite (see Figure S6) and indeed found comparable acetate accumulation matched lactate utilization between these materials. Lactate consumption occurs along with Fe(III) reduction, but a small amount of Fe(III) reduction also occurs when lactate is absent (Figure 3b). We attribute this to residual hydrogen present in the anaerobic chamber, which could also explain our observations for Cr(VI) cycling at long timescales (i.e., after lactate is consumed, hydrogen can still provide reducing equivalents). We have commented on this in the revised results and discussion sections (see Lines 321-327 and Lines 381-391).

Based on Fig. 2, growth ceases after 12 hours, yet most of Fe(II) is released (for MIL-100) after that time point. This finding remains unexplained. It is also difficult to understand how the reduction of 0.5 mM Fe(III) is sufficient to support growth to a cell density of 10⁸ cells. Are we looking here at a dissolution kinetics?

Even after cell growth ceases at 12 h, it is likely that respiration is still required for cell maintenance (Saville et al. *J. Bacteriol.* 2011, Rowe et al. *mBio* 2018) and indeed our iron reduction kinetics appear constant and linear past this time. Similarly, others have observed continued soluble Fe(III) reduction well beyond plateauing of MR-1 growth (Myers & Myers *J. Appl. Bacteriol.* 1994). Our iron reduction trends are probably similar during early cell growth, but it is challenging to obtain accurate measurements for both biomass and Fe(II) concentration at these early times using our assays (Bradford and Ferrozine). We attempted to use more sensitive detection assays, but found that the MOFs interfered with these due to high background caused by non-specific adsorption. Overall, our growth results are consistent with previous reports showing similar growth yields regardless of electron acceptor (Kostka et al. *Appl. Environ. Microbiol.* 2002). Given this, iron reduction does not appear to be growth limiting under our conditions, as suggested by CFU counts and protein content reaching similar levels across MIL-100, Fe-BTC, and ferrihydrite. While we acknowledge our cell densities appear high given the Fe(II) levels measured, we note that few studies have explicitly measured *Shewanella* growth yields concomitantly with Fe(II) generation (Myers & Myers *J. Appl. Bacteriol.* 1994, Liu et al. *Biotechnol. Bioeng.* 2001) with fine time course resolution. Moreover, since Fe(III) reduction continues past stationary-phase, comparisons between reported cell/Fe(II) yields and our results are likely not analogous. Interestingly, our estimated growth yield (2x10¹¹ cells/mmol e transferred to Fe) compare favorably to growth yields calculated from the initial report of *S. oneidensis* MR-1 (1.6 to 8x10¹⁰ cells/mmol e transferred to Mn, Myers & Nealson *Science* 1988). Per our metabolite measurements, the stationary phase cells still appear metabolically active given their ability to reduce Fe(III), even after Cr(VI) addition. While substantial amounts of Fe(II) are in the MOF after 48 h (ca. 50%, see Figure 4), dissolution kinetics could play a role and we speculate this may be one reason why the MOFs show higher reduction rates.

Biological rates should be given as specific rates and normalized to the biomass (Fig. 3). Also based on the data of Fig. 3c, the involvement of flavins is slight, and the statement in the following sentence that ‘...flavins are important contributors to Fe(III) reduction...’ is not warranted (176-181).

We have repeated all reduction rate experiments and measured the endpoint biomass (total protein content via Bradford) for these cultures. We have normalized all reduction rates by biomass and changed the results and discussion on the role of flavins according per the Reviewer’s suggestions (see Lines 179-186 and Lines 332-333). Indeed, when normalized to biomass, the role of flavins for MOF reduction appears unimportant (see Figure 3c and Figure S9).

It is argued (Fig. S7 and 152-158) that *S. oneidensis* can reduce the Fe(III)-organic framework, which leads to a significant but not complete dissolution of the framework complex. How then, can the framework stability be maintained if Fe(II) partially dissolves? How is that ‘stability’ determined?

It is unclear how stability can be maintained in the frameworks except if Fe(III/II) is in equilibrium between the framework and solution. An alternative explanation may be that smaller particles degrade while larger particles remain intact. We revised Figure 4 to show that Fe(III) reduction takes place within the MOF structure (as evidenced by color change) as well as through Fe solubilization. Stability was determined by PXRD post-reduction. If MOFs are unstable, their PXRD pattern should change or peaks should disappear, as is the case with MIL-88A. MIL-100 and Fe-BTC have patterns similar to their as-synthesized ones, indicating they have not degraded. Furthermore, our post-reduction SEM and TEM images show that insoluble material is still present (degradation should result in dissolution of the material). In the case of the Cr(VI) removal experiments, there is an isostructural version of MIL-100 that contains Cr(III) instead of Fe(III). Thus, it is possible that Cr(III) replaces Fe(III) in the original MIL-100 framework and improves framework stability once Fe(III) is removed. Unfortunately, we could not verify this, but suggest it as a possibility.

Cr(VI) reduction experiments. These experiments are very interesting! The experiments were conducted with 24 hrs old cultures. What was the state of metabolic activity in these cultures? Was there any lactate left at 24hrs or Fe(III) reduction activity left? Was the Fe(III) reduction rate observed due to microbial activity or the kinetics of Fe(II) dissolution (203)?

What mechanism do the authors propose to explain why no corresponding decrease in Cr(VI) was observed, despite appreciable concentration of Fe(II) in the ferrihydrite experiment (Fig.4)? Please discuss in text.

We have addressed these questions in the revised discussion. We observe continuous Fe(III) reduction activity after each Cr(VI) addition (Figure S16), suggesting cells are still metabolically active. In other experiments, lactate is ~half consumed after 48 hours. For long term cultures, such as the Cr(VI) cycling experiment, we suggest that hydrogen (from the anaerobic chamber atmosphere) could be acting as an additional electron donor. The MOFs alone show almost no Fe(II/III) leaching (Figure S1) so Fe(III) reduction/leaching is due to microbial activity. Finally, we added more discussion and a potential mechanistic explanation for why MOFs show superior Cr(VI) reduction and adsorption (see Lines 368-381).

Discussion

The discussion is lengthy and redundant to a large part. Again here, the emphasis is on metal-organic frameworks in general rather than on the issues identified in this study. The ‘advantage of these materials is their high degree of synthetic tunability’ was emphasized, but little systematic, mechanistic studies for a better fundamental understanding how these diverse structures explain different microbial reduction rates was provided. Furthermore, significant space (282-311) was devoted to elaborate on the potential of microbial-metal-organic framework systems, including the limitations on stability in water. This is interesting but unrelated to the data presented. The authors should have presented experimental data that address these points. In the absence of that, these sections read more like an advertisement for metal-organic frameworks rather than a critical discussion of the data presented.

We agree with the reviewer and have altered the discussion to provide a more mechanistic discussion on our observations, particularly for the increased Fe(III) reduction rates observed in MOFs as well as their superior Cr(VI) adsorption (relative to ferrihydrite).

The mode of toxicity of MIL-88A should be discussed.

We agree with the reviewer and this has been added to the discussion (see Lines 358-367).

Editorial Note: Reviewer #3 was unable to review the revision so a new reviewer #4 was invited to assess the authors responses to reviewer #3.

Reviewers' comments:

Reviewer #1 (Remarks to the Author):

Authors have answered the questions raised by me in a satisfactory manner. Hence the manuscript can be accepted in the current form

Reviewer #2 (Remarks to the Author):

This is a second look at the paper, and it is vastly improved. In my opinion, the questions raised by the reviewers have been addressed properly.

A few minor points:

Line 31: This sentence is only partly true. There are many metal-reducing and -oxidizing bacteria that are NOT electroactive (producing sulfide or other reductants, or producing oxidizing agents like peroxide, or even just altering the pH with the excretion of organics like pyruvate). Safer to say: .. support the growth and/or metabolism of many EET-capable microorganisms ... (it is a small point, but an important one, as metal transformations can be made by many different mechanisms).

Line 81: This is not a minimal medium – it has 0.5 g (500 mg) CAA (undefined amino acid mix), which MR-1 is known to use as both a C & N source. Also, I assume that the HEPES buffer is 100 mM (please change).

Line 126: Is the Bradford assay referenced somewhere? Good to have it here. (these days many students have never done this technique!!).

Good job with the riboflavin experiments! Using the lower levels is the way to do it.

Reviewer #4 (Remarks to the Author):

Overall the authors have done a nice job responding to the first round of reviewer comments. Upon reading the revised manuscript (I was not an initial reviewer) I have a few other minor points to raise for consideration.

Line 88 – what is the detection limit of the method used to quantify fumarate? Note *S. oneidensis* is able to use fumarate as an electron acceptor for anaerobic growth quite robustly, which can shift the dissolution equilibrium significantly for MIL-88A. Given the apparent toxicity, this point doesn't seem to be super relevant, but clarification on detection limit would be helpful.

Line 102 the authors claim that CFU of MR-1 in acceptor-free medium is constant over a 24h timescale. The claim appears to discount the near order of magnitude increase in CFU in Figure S3. Does this growth reflect carryover of fumarate, contamination of oxygen or something else? The authors should be cautious making inferences about growth with so few timepoints assayed. The culture may have grown significantly, entered stationary phase and died back – a single CFU timepoint could miss important growth dynamics. I am not suggesting additional experiments, just that the authors acknowledge the limitation of endpoint-type assays.

Line 104: Is *E. coli* capable of lactate fermentation or was fumarate also added to these cultures? Might this suggest some oxygen in the system that is being respired?

Line 107: to better test toxicity, was growth with fumarate impaired when MIL-88A was added? Just curious, not essential.

Line 187: the authors may wish to make a distinction between the role of flavins and reduction of insoluble iron oxides vs. chelated iron(III) substrates where flavins have been shown to enhance reduction rates of insoluble but not soluble. The MOF observation may be related to surface area / accessibility that is more like a soluble substrate than iron oxide. Moreover, the observation that flavin addition did not enhance MOF reduction rates would be consistent with 'accessible' iron being soluble (rather than insoluble). Did flavin addition enhance ferrihydrite reduction, as has been previously observed? Was the bfe mutant strain defective in ferrihydrite reduction?

Line 424: lyophilization may alter the crystallinity and/or size of ferrihydrite, decreasing the rate at which *S. oneidensis* can use the substrate. Typically the protocol used is driven from U Schwertmann, RM Cornell, 2000.

Missing italics throughout the references

Reviewers' comments (black) and authors' responses (blue):

Reviewer #1 (Remarks to the Author):

Authors have answered the questions raised by me in a satisfactory manner. Hence the manuscript can be accepted in the current form

We thank the reviewer for their helpful comments that have improved the manuscript.

Reviewer #2 (Remarks to the Author):

This is a second look at the paper, and it is vastly improved. In my opinion, the questions raised by the reviewers have been addressed properly.

A few minor points:

Line 31: This sentence is only partly true. There are many metal-reducing and -oxidizing bacteria that are NOT electroactive (producing sulfide or other reductants, or producing oxidizing agents like peroxide, or even just altering the pH with the excretion of organics like pyruvate). Safer to say: .. support the growth and/or metabolism of many EET-capable microorganisms ... (it is a small point, but an important one, as metal transformations can be made by many different mechanisms).

We apologize for this and thank the reviewer for the suggestion in wording. This sentence has been revised to reflect this distinction.

Line 81: This is not a minimal medium – it has 0.5 g (500 mg) CAA (undefined amino acid mix), which MR-1 is known to use as both a C & N source. Also, I assume that the HEPES buffer is 100 mM (please change).

We thank the reviewer for bringing this to our attention and have removed references to the medium as minimal (Line 157). We also thank the reviewer for pointing out our mistake in reporting the concentration of HEPES buffer. It is 100 mM, not 100 M, and this has been corrected in the text (Line 82).

Line 126: Is the Bradford assay referenced somewhere? Good to have it here. (these days many students have never done this technique!!).

Our method of performing the Bradford assay technique is detailed in the Materials and Methods section, specifically in Lines 463-470. We have included the original Bradford assay reference in line 468.

Good job with the riboflavin experiments! Using the lower levels is the way to do it.

Reviewer #4 (Remarks to the Author):

Overall the authors have done a nice job responding to the first round of reviewer comments. Upon reading the revised manuscript (I was not an initial reviewer) I have a few other minor points to raise for consideration.

Line 88 – what is the detection limit of the method used to quantify fumarate? Note *S. oneidensis* is able to use fumarate as an electron acceptor for anaerobic growth quite robustly, which can shift the dissolution equilibrium significantly for MIL-88A. Given the apparent toxicity, this point doesn't seem to be super relevant, but clarification on detection limit would be helpful.

Based on the signal to noise ratio of the NMR and probe used (400 MHz with OneNMR probe), we calculated our limit of detection to be roughly 18.6 μM . We have included mention of this in the caption for the NMR figure (Figure S1).

Line 102 the authors claim that CFU of MR-1 in acceptor-free medium is constant over a 24h timescale. The claim appears to discount the near order of magnitude increase in CFU in Figure S3. Does this growth reflect carryover of fumarate, contamination of oxygen or something else? The authors should be cautious making inferences about growth with so few timepoints assayed. The culture may have grown significantly, entered stationary phase and died back – a single CFU timepoint could miss important growth dynamics. I am not suggesting additional experiments, just that the authors acknowledge the limitation of endpoint-type assays.

We thank the reviewer for their comments and have adjusted the language of Lines 103-105 to reflect that CFU count is slightly increasing. Because the cells have been washed twice and diluted with fumarate-free medium, it is unlikely that the carryover of fumarate is significant. Additionally, all of our solutions were degassed and stored anaerobically in a chamber with constant monitoring of the O_2 concentration. However, as mentioned by Reviewer 2, casamino

acids may serve as an appreciable nutrient source for *S. oneidensis*. In the original *S. oneidensis* MR-1 report (Myers & Nealson *Science* 1988) it was found that *S. oneidensis* can utilize glycine as an electron acceptor. As glycine is found in casamino acids (based on the 0.05% casamino acids in our medium formulation and literature composition values, we estimate it to be ~77 μ M, Nolan *Mycologia* 1971), it may be able to support the modest increase in cell counts. While we do not wish to speculate further on the role of glycine/casamino acids, we have addressed the limitations of the assay in the Lines 369-371 of the discussion.

Line 104: Is *E. coli* capable of lactate fermentation or was fumarate also added to these cultures? Might this suggest some oxygen in the system that is being respired?

Previous observations in our lab have shown that *E. coli* is unable to grow on lactate and fumarate in Hungate tubes with an N₂ atmosphere. However, in the same medium, they have been able to grow in an anaerobic chamber with and without fumarate when lactate is present. Given that the atmosphere of the chamber includes 3% H₂ and *E. coli* has been shown to undergo hydrogen-stimulated growth on other non-fermentable carbon sources (Sawyers *Antonie Leeuwenhoek* 1994, Macy et al. *J. Bacteriol.* 1976), it is likely this process accounts for the increase in cell density.

Line 107: to better test toxicity, was growth with fumarate impaired when MIL-88A was added? Just curious, not essential.

We thank the reviewer for this suggestion and agree that this would be an interesting experiment to test whether extra respiratory power alleviates MIL-88A toxicity. However, since we observed no decrease in cell density with acceptor-free medium but an extreme decline in cell density for MR-1/MIL-88A suspensions, we simply concluded that MIL-88A is toxic when it is the sole electron acceptor. We agree that other growth additives (different carbon sources/additional electron acceptors) may alter the degree of toxicity but feel that a systematic examination is best left for a separate manuscript.

Line 187: the authors may wish to make a distinction between the role of flavins and reduction of insoluble iron oxides vs. chelated iron(III) substrates where flavins have been shown to enhance reduction rates of insoluble but not soluble. The MOF observation may be related to surface area / accessibility that is more like a soluble substrate than iron oxide. Moreover, the observation that flavin addition did not enhance MOF reduction rates would be consistent with 'accessible' iron being soluble (rather than insoluble). Did flavin addition enhance ferrihydrite reduction, as has been previously observed? Was the bfe mutant strain defective in ferrihydrite reduction?

We thank the reviewer for noting this distinction and apologize if our discussion on this matter was not clear. We do interpret the flavin/bfe results as indication that solubilized Fe(III) is the primary contributor to fast MOF reduction and have already briefly mentioned this in the discussion (lines 333-336). However, to further highlight the non-influence of flavins on soluble Fe(III) reduction, we have included mention of this in the results section that describes flavin/bfe experiments (Line 187). Although not shown, previous experiments in our lab have observed that the bfe mutant is defective in ferrihydrite reduction and flavin addition to MR-1 at non-physiological levels (10 micromolar) enhanced ferrihydrite reduction, but this did not occur at lower concentrations (1 micromolar). Since results from this manuscript and a recent report (Kees et al. *Appl. Environ. Microbiol.* 2019) have shown that flavin supplementation/deficiency may affect growth rate separate from Fe(III) reduction, we chose to exclude a deeper examination since their role in metal reduction is controversial and not fully understood.

Line 424: lyophilization may alter the crystallinity and/or size of ferrihydrite, decreasing the rate at which *S. oneidensis* can use the substrate. Typically the protocol used is driven from U Schwertmann, RM Cornell, 2000.

We agree with the reviewer that our method of ferrihydrite preparation has the potential to alter the crystallinity and reduction rate of the material relative to other protocols. Thus, we have mentioned in the discussion (Lines 330-332) that our measured reduction rates are subject to change based on ferrihydrite preparation. Despite the potential for mineralogical differences, our rates appear comparable to similar *Shewanella* studies (Coursolle & Gralnick, *Molecular Biology* 2010). With that said, it is difficult to make direct comparisons across the literature as differences in experimental setup (e.g., inoculating cell density, media components, container geometry, etc.) can have large effects on rate (Liu et al. *Environ. Sci. Technol.* 2001, Dippon et al. *Geomicrobiol. J.* 2015). We would also like to note that the variable performance of ferrihydrite based on processing conditions highlights an advantage of MOFs. The MOFs that are stable in water do not undergo thermodynamic phase changes, as detected by PXRD, akin to ferrihydrite and different methods of preparation still result in consistent crystallinity.

Missing italics throughout the references

We thank the reviewer for bringing this to our attention and note that the reference formatting has been corrected.

REVIEWERS' COMMENTS:

Reviewer #4 (Remarks to the Author):

Nothing new to add, the revised manuscript looks to be in good shape.